# What do larger image classifiers memorise?

**Michal Lukasik** *mlukasik@google.com*
*Google Research*

**Vaishnavh Nagarajan** *vaishnavh@google.com*
*Google Research*

**Ankit Singh Rawat** *ankitsrawat@google.com*
*Google Research*

**Aditya Krishna Menon** *adityakmenon@google.com*
*Google Research*

**Sanjiv Kumar** *sanjivk@google.com*
*Google Research*

**Reviewed on OpenReview:** *https://openreview.net/forum?id=Ew73inSyhG*

## Abstract

The success of modern neural networks has prompted study of the connection between *memorisation* and *generalisation*: overparameterised models generalise well, despite being able to perfectly fit ("memorise") completely random labels. To carefully study this issue, Feldman (2019) proposed a metric to quantify the degree of memorisation of individual training examples, and empirically computed the corresponding memorisation profile of a ResNet on image classification benchmarks. While an exciting first glimpse into what real-world models memorise, this leaves open a fundamental question: *do larger neural models memorise more?* This aligns with the common practice of training models of different sizes, each offering different cost-quality trade-offs: while larger models are typically observed to have higher quality, it is of interest to understand whether this is merely a consequence of them memorising larger numbers of input-output patterns. We present a comprehensive empirical analysis of this question on image classification benchmarks. We find that training examples exhibit an unexpectedly diverse set of memorisation trajectories across model sizes: most samples experience *decreased* memorisation under larger models, while the rest exhibit *cap-shaped* or *increasing* memorisation. We show that various proxies for the Feldman (2019) memorisation score fail to capture these fundamental trends. Lastly, we find that knowledge distillation — an effective and popular model compression technique — tends to inhibit memorisation, while also improving generalisation. Specifically, memorisation is mostly inhibited on examples with increasing memorisation trajectories, thus pointing at how distillation improves generalisation.

## 1 Introduction

Statistical learning is conventionally thought to involve a delicate balance between *memorisation* of training samples, and *generalisation* to test samples (Hastie et al., 2001). However, the success of *overparameterised* neural models challenges this view: such models generalise well, despite having the capacity to memorise, e.g., by perfectly fitting completely random labels (Zhang et al., 2017). Indeed, in practice, such models typically *interpolate* the training set, i.e., achieve zero misclassification error. This has prompted a series of analyses aiming to understand why such models can generalise (Bartlett et al., 2017; Brutzkus et al., 2018; Belkin et al., 2018; Neyshabur et al., 2019; Bartlett et al., 2020; Wang et al., 2021).

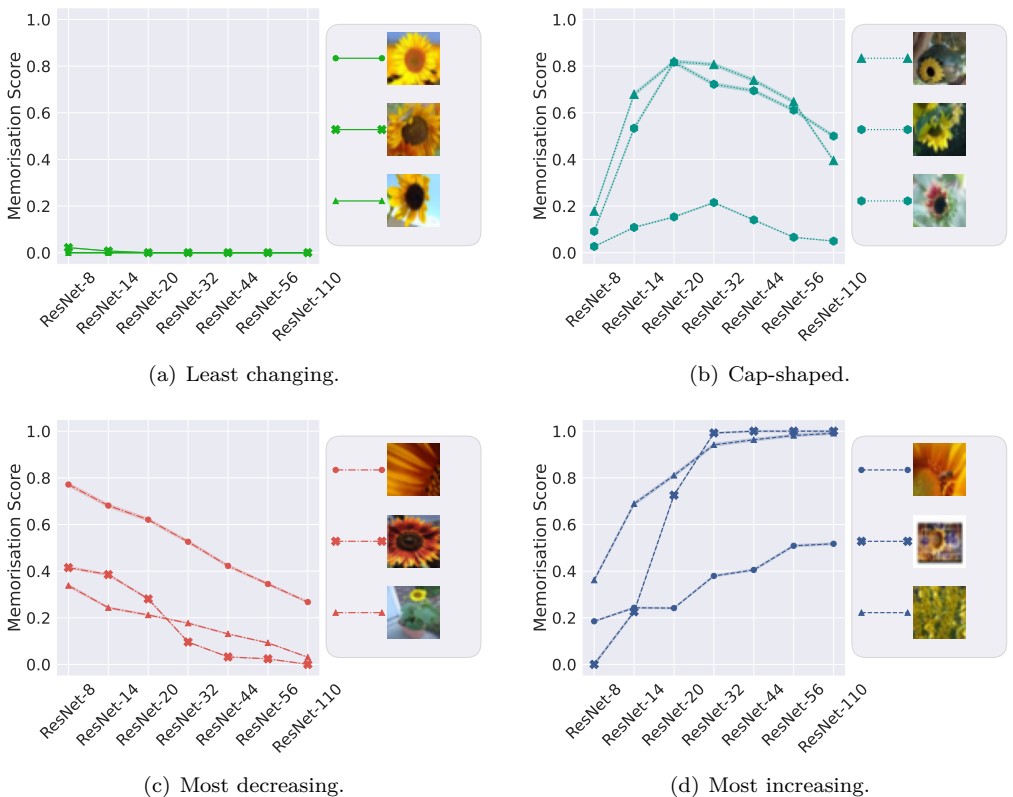

Figure 1: Training examples on CIFAR-100 *exhibit a diverse set of memorisation trajectories* across model depths: fixing attention on training examples belonging to the `sunflower` class, while many examples unsurprisingly have fixed, decreasing or cap-shaped memorisation scores (green, red and teal curves), there are also examples with increasing memorisation *even after interpolation* (blue curves). In §3.2 we discuss their characteristics in more detail, e.g. how these trajectories constitute a vast majority of trajectories seen in the data.

Recently, Feldman (2019) established that in some settings, memorisation may be *necessary* for generalisation. Here, "memorisation" is defined via a theoretically-grounded stability-based notion, where the high memorisation examples are the ones that the model can correctly classify only if they are present in the training set (see Equation 1 in §2). This definition allows the level of memorisation[1] of a training sample to be *estimated* for real-world neural models. To that end, Feldman & Zhang (2020) studied the memorisation profile of a ResNet on standard image classification benchmarks.

While an exciting first glimpse into how real-world models memorise, it does not tell us how memorisation *varies with* model size. After all, varying the model size is an important practical consideration: training models of different sizes within a single family (e.g., ResNet (He et al., 2016b), MobileNet (Howard et al., 2019b), and T5 (Raffel et al., 2020)) is commonplace, as it enables practitioners to deploy the best performing models while respecting their computation budget for both training and inference. Is the improved performance of larger models merely a result of them memorising a larger number of input-label patterns? While larger models have more *capacity* for memorisation, it is yet not well understood whether the result of the commonly used training recipes (e.g., based on SGD) indeed result in models that increasingly memorise more as the model size increases.

---

[1]From hereon in, unless otherwise noted, we shall use "memorisation" to refer to the stability-based notion of Feldman (2019); see Equation 1.

Practitioners also employ a systematic approach to obtain high quality models of varying size: knowledge distillation. In particular, it involves developing good-quality small (student) models by drawing supervision from high-performing large (teacher) models. Fundamentally, distillation involves interaction among models of different sizes, and thus a further question can be posed: how does the interaction between the teacher and the student impact the smaller model's memorisation?

This paper conducts a systematic and comprehensive empirical study to answer the aforementioned questions. In particular, using the theoretically-grounded notion of memorisation from Feldman (2019), we analyse the interplay between memorisation and generalisation across different model sizes. Based on our controlled experiments, we make the following contributions:

(i) we present a quantitative analysis of how memorisation varies varies with model complexity (e.g., depth or width of a ResNet) for image classifiers. Our main findings are that increasing the model complexity tends to make the distribution of memorisation across examples more *bi-modal* (Section 3.3). At the same time, we identify that existing computationally-tractable alternatives of quantifying memorisation and example difficulty do not capture this key trend (Section 3.5).

(ii) to further probe into the bi-modal memorisation trend, we present examples exhibiting varying memorisation score trajectories across model sizes, and identify four most prominent trajectory types, including those where memorisation *increases* with model complexity. We find that particularly *ambiguous and mislabeled* examples exhibit this kind of trajectory (Section 3.4).

(iii) we conclude with a quantitative analysis demonstrating that distillation tends to inhibit memorisation, particularly of samples that the one-hot (i.e., non-distilled) student memorises. Intriguingly, we find that the memorisation is mostly inhibited on the examples for which memorisation increases as the model size increases. This observation leads to conclude that distillation improves generalisation by limiting memorisation of such hard examples (Section 4).

## 2 Background and related work

The term "memorisation" is often invoked when discussing supervised learning, but with several distinct meanings. A classical usage refers to models that simply construct a fixed *lookup table*, *viz.* a table mapping certain keys to targets (e.g., labels) (Devroye et al., 1996; Chatterjee, 2018). Noting that the 1-nearest neighbour algorithm can *interpolate* the training data (i.e., perfectly predict every training sample), some works instead use "memorisation" as a synonym for interpolation (Zhang et al., 2017; Arpit et al., 2017; Stephenson et al., 2021). A more general definition is that "memorisation" occurs when the training error is lower than the best achievable error (or *Bayes-error*) (Bubeck et al., 2020; Cheng et al., 2022). We summarise more notions of memorisation in Table 1.

While each of these notions imply "memorising" the *entire* training set, one may also ask whether a *specific* sample is memorised. One intuitive formalisation of this notion is that the model predictions change significantly when a sample is removed from training (Feldman, 2019; Jiang et al., 2021b), akin to algorithmic stability (Bousquet & Elisseeff, 2002). More precisely, consider a training sample $S$ comprising $N$ *i.i.d.* draws from some distribution $D$ over labelled inputs $(x, y) \in \mathcal{X} \times \mathcal{Y}$. A learning *algorithm* is some randomised function $\mathsf{A}(\cdot; S) \colon \mathcal{X} \to \mathcal{Y}$, where the randomness is, e.g., owing to initialisation, ordering of mini-batches, and stochasticity in parameter updates. The *memorisation score* of a sample $(x, y) \in S$ is then (Feldman, 2019):

$$\mathsf{mem}(x, y; S) = \underbrace{\mathbb{P}(y = \mathsf{A}(x; S))}_{\text{in-sample acc.}} - \underbrace{\mathbb{P}(y = \mathsf{A}(x; S - \{(x, y)\}))}_{\text{out-sample acc.}}. \tag{1}$$

Here, $\mathbb{P}(\cdot)$ considers the randomness in the learning algorithm. Intuitively, this is the excess classification accuracy on the sample $(x, y)$ when it is *included* versus *excluded* in the training sample. Large neural models can typically drive the first term to 1 for *any* $(x, y) \in S$ (i.e., they can interpolate any training sample); however, some $(x, y)$ may be very hard to predict when they are not in the training sample. Such examples may be considered to be "memorised", as the model could not "generalise" to these examples based on the rest of the training data alone.

Despite its strengths, Equation 1 has an obvious drawback: it is prohibitive to compute in most practical settings. Indeed, it naïvely requires that we retrain our learner at least $N$ times, with *every* training sample excluded once; accounting for randomness in $\mathsf{A}$ requires further repetitions. Feldman & Zhang (2020) provided a more tractable estimator, wherein for fixed integer $M$, one draws $K$ *sub-samples* $\{S^{(k)}\}_{k\in[K]}$ uniformly from $\mathcal{P}_M(S)$, the set of all $M$-sized subsets of $S$. For fixed $(x,y)$, let $K_{\mathrm{in}} \doteq \{k \in [K]\colon (x,y) \in S^{(k)}\}$, and $K_{\mathrm{out}} \doteq \{k \in [K]\colon (x,y) \notin S^{(k)}\}$ denote the sub-samples including and excluding $(x,y)$, respectively. We then compute

$$\widehat{\mathsf{mem}}_{M,K}(x,y;S) = \frac{1}{|K_{\mathrm{in}}|}\sum_{k\in K_{\mathrm{in}}} [\![y = \mathsf{A}(x;S^{(k)})]\!] - \frac{1}{|K_{\mathrm{out}}|}\sum_{k\in K_{\mathrm{out}}} [\![y = \mathsf{A}(x;S^{(k)})]\!]. \tag{2}$$

This quantity estimates $\mathsf{mem}(x,y;S)$ to precision $\mathcal{O}(1/\sqrt{K})$ (Feldman & Zhang, 2020). Jiang et al. (2021a) considered a closely related quantity, namely *consistency score* (or *C-score*), defined as $\mathsf{cscore}(x,y;S) = \mathbb{E}_{M\sim\mu}\left[\mathbb{E}_{S'\sim\mathcal{P}_M(S)}\left[\mathbb{P}(y = \mathsf{A}(x;S' - \{(x,y)\}))\right]\right]$, where $\mu$ is a distribution over integers. When $\mu(\cdot)$ is a point-mass, this is the second term in $\widehat{\mathsf{mem}}_{M,K}(x,y;S)$. Note that the first term in $\widehat{\mathsf{mem}}_{M,K}(x,y;S)$ is typically 1 for overparameterised models, since they are capable of interpolation. Jiang et al. (2021a) also proposed effective *proxies* for the C-score, which rely on the model behaviour across training steps. Suppose we have a model that is iteratively trained for steps $t \in \{1,2,\ldots,T\}$, where at $t$-th step, the model produces a probability distribution $\hat{p}^{(t)}\colon \mathcal{X} \to \Delta(\mathcal{Y})$ over the labels. Jiang et al. (2021a) argued that the metric capturing the temporal average of the probability assigned to the true label:

$$\mathsf{cprox}(x,y;S) = \mathbb{E}_t\left[\hat{p}_y^{(t)}(x)\right] \tag{3}$$

can correlate strongly with (the point mass version of) $\mathsf{cscore}(x,y;S)$.

**Relating memorisation and example difficulty.** Equation 3 provides an interesting bridge between memorisation and *example difficulty*. For example, TracIn (Pruthi et al., 2020) and GRAD (Paul et al., 2021) also use the evolution of model predictions across training steps to identify samples that are difficult to learn; roughly, these are samples which cause large loss updates when they are trained on. Equation 3 is also related to the notion of *forgetting* (Toneva et al., 2019; Zhou et al., 2022; Maini et al., 2022): a count of the number of transitions from *learned* to *forgotten* during training an example undergoes. Another related metric is the *learning speed*: the earliest training iteration after which the model predicts the ground truth class for that example in all subsequent iterations. A different notion of difficulty introduced by Nohyun et al. (2023) is the *CG score*, which relies on calculating the gap in generalization error bounds of an overparameterized two-layer network with ReLU activations when an example is *excluded* from training. Another related measure of sample difficulty is the RHO-loss (Mindermann et al., 2022), wherein the training loss is contrast with the *irreducible* loss when a sample is only present in a holdout set. The C-score also correlates with the *prediction depth* (Baldock et al., 2021), which computes model predictions at intermediate layers, and reports the earliest layer beyond which all predictions are consistent.

In a related line of work, Ghorbani & Zou (2019) proposed the *data Shapley score* to capture the value of a training example with respect to a train dataset, a learning algorithm and an evaluation metric. The score differs from the memorisation score in that the impact of an example is evaluated with respect to *all* subsets of the training set, as opposed to the entire training set. Interestingly, this is similar to Equation 2, which samples fixed-size subsets of the training set.

**Memorisation versus generalisation**. Given that the ultimate goal of statistical learning is *generalisation*, it is natural to ask whether this is at odds with "memorisation". A classical result establishes that a lookup table as implemented by the $k$-NN algorithm is universally consistent (Stone, 1977). Interpolating models such as boosting with decision stumps have similarly been shown to generalise (Bartlett et al., 1998), and more refined analyses have been conducted for modern interpolating neural models (Bartlett et al., 2017; Dziugaite & Roy, 2017; Brutzkus et al., 2018; Belkin et al., 2018; Neyshabur et al., 2019; Liang & Rakhlin, 2020; Montanari & Zhong, 2020; Bartlett et al., 2020; Vapnik & Izmailov, 2021). Intriguingly, some recent works have established that under certain settings, "memorisation" may be *necessary* for generalisation, either in the sense of interpolation (Cheng et al., 2022), stability-based label memorisation (Feldman, 2019), or stronger example-level memorisation (Brown et al., 2021).

**Implicit versus explicit memorisation**. Memorisation has received particular interest in the context of large language models (LLMs), such as GPT (Brown et al., 2020) and T5 (Raffel et al., 2020). Here, "memorisation" typically refers to the ability of a model to recall factual information present in the training set (e.g., names of individuals) (Petroni et al., 2019; Roberts et al., 2020; Carlini et al., 2021; 2022). This aligns with the notion from statistical learning of "memorisation" as employing a lookup table. While LLMs are capable of *implicit* memorisation, several works have shown benefits from augmenting neural models with an *explicit* memorisation component (Lample et al., 2019; Guu et al., 2020; Khandelwal et al., 2020; Borgeaud et al., 2021; Yang et al., 2023). Similar ideas have also proven useful outside of NLP (Panigrahy et al., 2021; Vapnik & Izmailov, 2021; Wang & Shao, 2022).

| Definition | References |
|---|---|
| Zero training error | (Zhang et al., 2017) |
| Zero training error + label is random | (Arpit et al., 2017; Yao et al., 2020; Stephenson et al., 2021) |
| Training error below Bayes error rate | (Bubeck et al., 2020; Cheng et al., 2022) |
| Prediction based on spurious correlations | (Sagawa et al., 2020; Glasgow et al., 2022) |
| Ability to reconstruct from other training samples | (Radhakrishnan et al., 2020; Carlini et al., 2021) |
| Inability to predict when removed from training sample | (Feldman, 2019; Jiang et al., 2021a) |

Table 1: Summary of existing definitions of "memorisation" of a training sample. In this paper, we focus on the final row, which proposes a stability-based metric quantifying sample predictability.

**Prior empirical analyses of memorisation**. Several works have studied the *interpolation* behaviour of neural models as one varies model complexity (Zhang et al., 2017; Neyshabur et al., 2019). Empirical studies of memorisation in the sense of fitting to random (noisy) labels was conducted in (Arpit et al., 2017; Gu & Tresp, 2019). These works demonstrated that real-world networks tend to fit "easy" samples first, and exhibit qualitatively different learning trajectories when presented with clean versus noisy samples. Zhang et al. (2020) provided an elegant study of the interplay between memorisation and generalisation for a regression problem, involving learning either a constant or identity function. Feldman & Zhang (2020) studied memorisation in the sense of the stability-based memorisation score in Feldman (2019) (cf. Equation 1), by quantifying the influence of each training example on different test examples. Based on these, they identified a subset of test examples for which the model significantly relies on the memorised (in the stability sense) training examples to make correct predictions. While the direct inspiration for our study, these experiments were for a single architecture on CIFAR-100 and ImageNet. Furthermore, they did not consider the impact of model distillation.

Concurrent work (Dam et al., 2023) analyses the effect of model compression specifically for CIFAR-10 and concur with our findings that compression inhibits memorisation. However, like Feldman (2019), most other works study the memorisation behavior individually for each model rather than consolidate trends across a family of models. Han et al. (2022) examine what images are typically memorised by a large variety of image classifiers. A line of work (Baldock et al., 2021; Maini et al., 2023; Stephenson et al., 2021) has investigated how to localize where an example is memorised in a given network. Other works have looked at the effect of orthogonal factors such as initialization (Mehta et al., 2021) on memorisation. Xu et al. (2023) examine the interplay between memorisation and robustness specifically for adversarially-trained models, while we study standard-trained models.

An orthogonal line of work has extended the concept of memorisation to language modeling. Zheng & Jiang (2022); Zhang et al. (2021) extend the formulation of memorisation in Feldman (2019) and confirm that the long tail theory holds in language datasets as well. However, most work in language models (Carlini et al., 2023; Bai et al., 2021; Biderman et al., 2023) take a qualitatively different approach to memorisation, wherein a model is said to memorise a datapoint if it can complete a prefix in the same way it was completed in the training set.

# 3 The unexpected tale of memorisation

To demystify the excellent performance of modern neural networks, a key step is developing a systematic understanding of their fundamental properties as we scale their capacity. It is known that as the model size grows, both the *interpolation* and *generalisation* of the model increase. Given that memorisation is a fundamental related property, it is natural to ask how the memorisation behavior of modern networks evolves with increasing model sizes. In this section, we present a detailed empirical study in this direction, while highlighting various nuanced and surprising observations. Before discussing our key findings, we begin by introducing the exact setup and scope of our empirical study.

## 3.1 Setup and scope

Quantifying the nature of memorisation requires picking a suitable definition of the term. Owing to its conceptual simplicity and intuitive alignment with the term "memorisation", we employ the stability based memorisation score of Feldman (2019), per Equation 1. For computational tractability, we employ the approximation to this score from Feldman & Zhang (2020), per Equation 2. This reduces the computational burden of estimating $\mathsf{mem}(x, y; S)$, but does not eliminate it: for *each* setting of interest, we need to draw a number of independent data sub-samples, and train a fresh model on each. This necessitates a tradeoff between the breadth of results across settings, and the precision of the memorisation scores estimated for any individual result. We favour the former, and estimate $\widehat{\mathsf{mem}}_{M,K}(x, y; S)$ via $K = 400$ draws of sub-samples $S' \sim \mathcal{P}_M(S)$, with $M = \lceil 0.7N \rceil$.

With this setup, we empirically examine a simple question: *how is memorisation influenced by model capacity?* Specifically, for a range of standard image classification datasets — CIFAR-10, CIFAR-100, and Tiny-ImageNet — we empirically quantify the memorisation score as we vary the capacity of standard neural models, based on the ResNet (He et al., 2016b;c) and MobileNet-v3 (Howard et al., 2019b) family (cf. Appendix D for precise settings). Here, it is worth highlighting that while Feldman (2019) studied memorisation profiles for fixed models, the *change* in memorisation behavior across model sizes has not been systematically studied before.

In the rest of the section, we present key findings from our empirical study, progressively exploring memorisation behavior at a more granular level. We start with discussing the average memorisation scores for networks and then present the distribution of memorisation scores across training examples for different sized models. We next explore per-example memorisation trajectory across model sizes which leads an intuitive categorization of all training examples into four categories. We conclude with inspecting whether the properties of memorisation we uncover hold under example difficulty metrics which were shown in previous works to highly correlate with the stability based memorisation.

## 3.2 Sufficiently large models memorise less on average

Previous works have claimed that larger networks imply more memorisation, albeit based on different notion of memorisation than ours (Zhang et al., 2017; Neyshabur et al., 2019; Carlini et al., 2023). This leads one to hypothesise that average memorisation score across training set, per Equation 1 or 2, should increase with model size. We now assess whether this hypothesis is borne out empirically.

In Figure 2, we visualise how ResNet model depth influences the memorisation score (Equation 2) on CIFAR-100 and ImageNet. Specifically, for each model, we report the *average* memorisation score across all training samples as a coarse summary. For CIFAR-100 we see that this score increases up to depth 20, and then, contrary to the naïve hypothesis above, starts *decreasing* (albeit slightly).

This (initially) puzzling phenomenon may be intuitively understood by breaking down the two terms used to compute the memorisation score in equation 2: the *in-sample accuracy*, and the *out-of-sample accuracy*. As expected, both quantities steadily increase with model depth. The increasing memorisation score up to depth 20 can be explained by the *in-sample accuracy* increasing *faster* than the *out-of-sample accuracy* up to this point; beyond this point, the in-sample accuracy saturates at 100%, while the out-of-sample accuracy keeps increasing. Thus, necessarily, the memorisation score starts to drop. In Appendix H.3, we

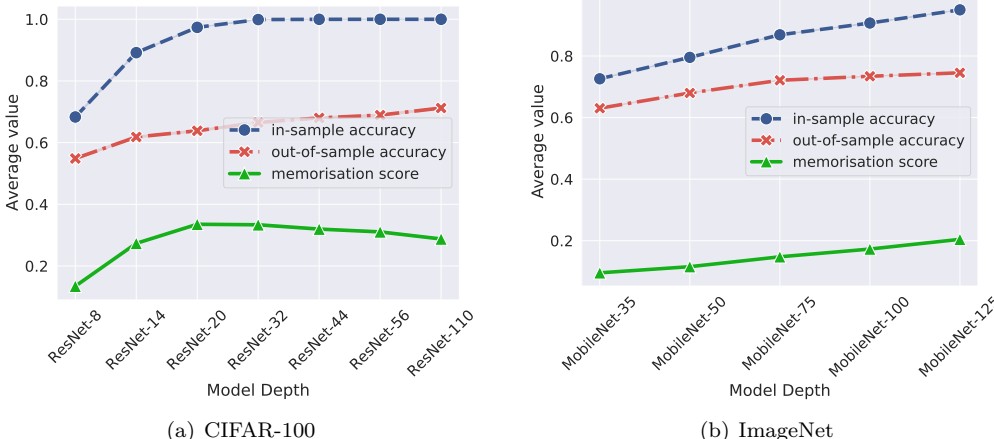

(a) CIFAR-100          (b) ImageNet

Figure 2: Average memorisation scores from models of varying size trained on CIFAR-100 (left plot) and ImageNet (right plot) training samples. On CIFAR-100, the average memorisation score steadily *increases* with depth up till the ResNet-20, and steadily *decreases* afterwards. The increase up to a certain depth can be explained by the in-sample accuracy increasing faster than out-of-sample accuracy (right plot), until the point where the former is $\sim 100\%$. On ImageNet, the in-sample accuracy does not reach $\sim 100\%$ and in effect the memorisation scores is seen as increasing.

give an alternative explanation for why memorisation decreases after interpolation. For ImageNet, we see memorisation steadily increasing due to in-sample accuracy increasing faster than out-of-sample accuracy, while the in-sample accuracy does not saturate.

### 3.3  Large models have increasingly bi-modal memorisation score distributions

The result of memorisation decreasing with an increasing model size discussed in Section 3.2 only considered the *average* memorisation score for a given model. But how does the *distribution* of memorisation scores vary with model capacity? To this end, Figure 3 plots precisely this distribution for each model in consideration. We observe that the memorisation scores tend to be *bi-modal*, with most samples' score being closer to 0 or 1. This is in agreement with observations made by Feldman & Zhang (2020) for a *fixed* architecture: ResNet on CIFAR-100 and MobileNet on ImageNet.

More interestingly, our finding is that this bi-modality is *exaggerated with model depth*: larger models have a higher fraction of samples with both memorisation score close to 0 and 1. The increase of samples with high memorisation score denotes an increasing generalisation gap on a subset of training points. We find this to be true across various datasets and architectures, as shown in Figure 6 (Appendix). While the increasing fraction of samples with low memorisation score is consistent with the finding on CIFAR-100 about the average memorisation score decreasing, it is surprising that there is also a subset of points with *increasing* memorisation.

### 3.4  On the diversity of memorisation trajectories over model sizes

The bi-modality of the memorisation scores that we observed in Section 3.3 suggests that there exist at least two kinds of examples: those whose memorisation scores increase with depth, and those whose scores decrease. Next, we seek to more carefully characterize what these examples are by analysing the trajectory of memorisation score for *individual* examples.

**Categorizing examples by their memorisation trajectories.**  In Figure 1, we report the average memorisation score as a function of model depth (size) for individual examples from the `sunflower` class of CIFAR-100 (see Appendix for presentation over more classes and across different classes, upholding our

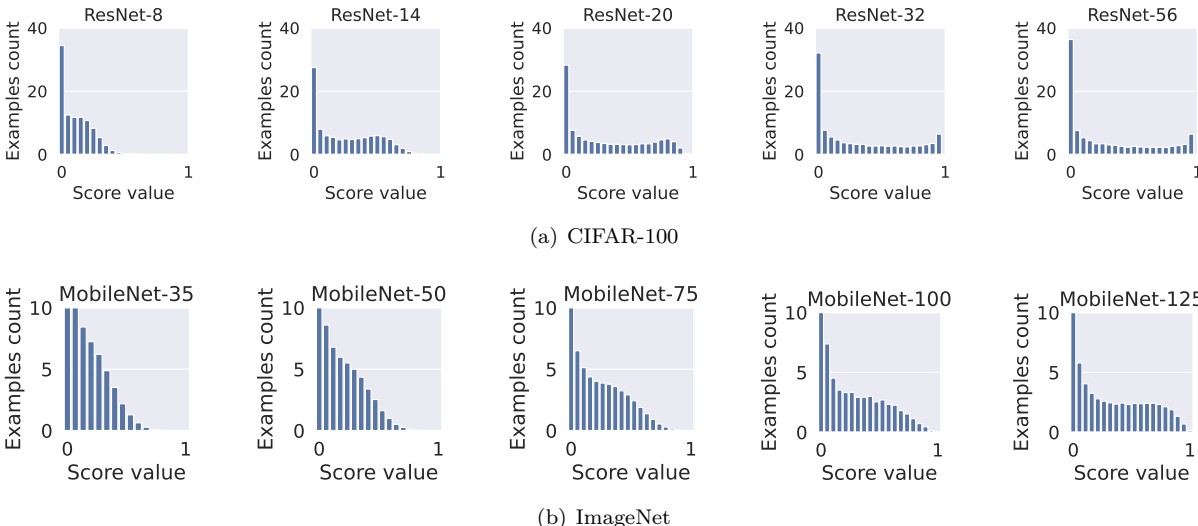

Figure 3: Distributions of memorisation scores from models trained on CIFAR-100 (top row) and ImageNet (bottom row) training samples. As model size increases, relatively more samples are highly memorised (notice the hump in the graphs from the top row is being progressively pushed to the right); Further, the number of examples with very low memorisation increases as well. Overall, we find a growing *bi-modality* of the distribution of memorisation scores with increasing size. In Figure 6 (Appendix). we show that the bi-modality of the memorisation score holds across more datasets (CIFAR-10, CIFAR-100, ImageNet, TinyImageNet) and model families (ResNet, MobileNet) and in Figure 7 we show this when varying model widths (as opposed to depths) to increase model sizes.

observations, as well as from the ImageNet models). We find four different kinds of trajectory patterns: increasing, decreasing, cap-shaped and constant. Also, in Table 12 (Appendix) we report predictions for the shown examples.

In Table 2 we report counts of examples in the CIFAR-100 data broken down by the trajectory they exhibit across model sizes. For robust counting, we treat a change in memorisation by less than 0.1 as a no change. We can see how the four trajectory types we present cover the majority of example trajectories.

**Interpreting the example trajectories.** We next offer a qualitative characterisation of the samples from the four trajectory types we identified. Such qualitative grouping of samples is common in work studying *example difficulty* Baldock et al. (2021); Jiang et al. (2021a); Feldman & Zhang (2020). We first notice that the least-changing memorisation examples are *easy and unambigous*. Quantitatively as well, we find that amongst these

| $\alpha$ | constant | increasing | decreasing | cap-shaped | other |
|---|---|---|---|---|---|
| 0.05 | 24.3% | 17.5% | 14.5% | 31.6% | 12.1% |
| 0.10 | 39.5% | 29.8% | 10.5% | 18.7% | 1.6% |

Table 2: Percentages of examples in the CIFAR-100 data as broken down by the trajectory type. For robust counting, we treat a change in memorisation by less than $\alpha$ as a no change.

examples, the peak memorisation score tends to be low (i.e., such samples tend to consistently not be memorised), suggesting that these are easy and unambiguous points. This phenomenon is also evident in Figure 9 (Appendix), where the examples with the least changing memorisation score are also least memorised scores by individual models. Next, we note that the examples corresponding to *increasing*, *cap-shaped* and *decreasing* memorisation trajectories are hard and ambiguous (cf. Figure 1). The *decreasing* and *cap-shaped* examples are arguably hard but with the ground-truth label correctly assigned in the data. On the other hand, the *increasing* examples are often multi-labeled (e.g., the first *increasing* example containing a bee) or mislabeled (e.g. the third *increasing* example), such that a human rater could be unlikely to label these images with the dataset provided label. This parallels the predictions from ResNet-110 for these examples,

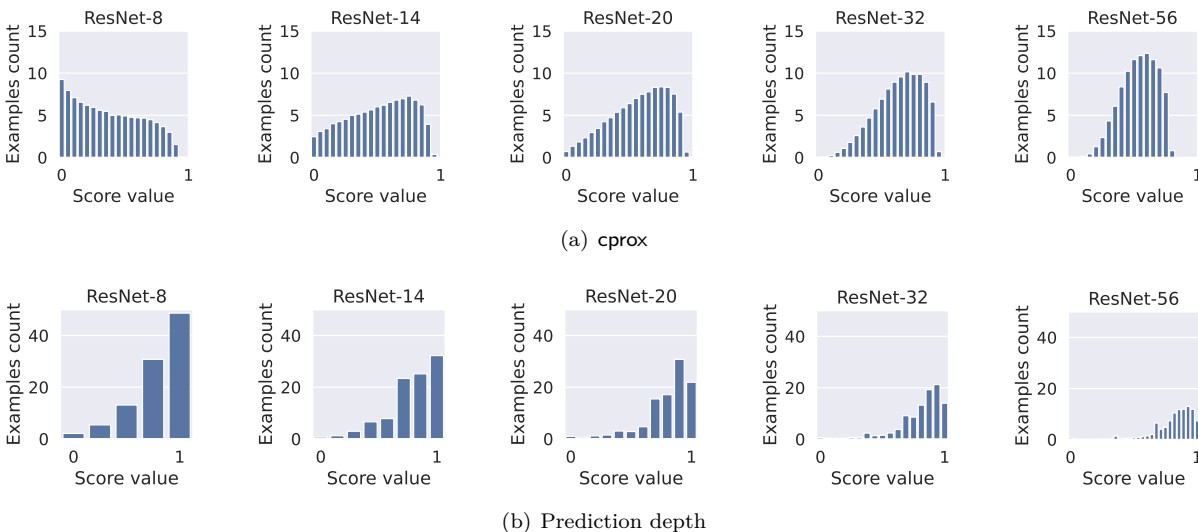

(a) cprox

(b) Prediction depth

Figure 4: Distributions of C-proxy scores (top row; see Equation (3)) and prediction depth (bottom row; see Baldock et al. (2021) and Appendix J for the details of our implementation) from models trained on CIFAR-100 training samples. Although these have high correlation with memorisation scores, as promised in prior work, they fail to capture the bi-modal behavior of memorisation.

which are arguably no less reasonable than the label as presented in the dataset, as can be observed in Table 12 (Appendix). We inspect more examples and further elaborate on our observations in Appendix H.

### 3.5 Do memorisation score proxies exhibit the same trends?

As discussed in Section 2, the stability-based memorisation score is difficult to compute. The C-score proxy cprox (Equation 3) was proposed in Jiang et al. (2021a) as a computationally efficient alternative to the C-score, a metric closely related to the stability-based memorisation score of Equation 1. Indeed, Jiang et al. (2021a) found this measure to have high *correlation* with the C-score, while cautioning that it should not be interpreted as an *approximation* to more fine-grained characteristics of the latter.

Going beyond correlation, however, we find in Figure 4 that the distribution of cprox scores have markedly different characteristics to stability-based memorisation: the former are *unimodal*, with most samples having a high score value. Recall that by contrast, stability-based memorisation scores exhibit a bi-modal distribution, with this phenomenon exaggerated with increasing model size (see Figure 3). Consequently, we do not observe many of the unexpected trends exhibited by stability-based memorisation, e.g., the possibility of cprox systematically *decreasing* for some samples as depth increases. We also report distributions from the example difficult metric called *prediction depth*, which has been shown to be closely related to the C-score, and analogously find that prediction depth yields a very different distribution than memorisation score (see Appendix).

Overall, we view the above discrepancy as an instance of the famous *Anscombe's quartet* (Anscombe, 1973) in statistics: it is possible for two distributions to have very similar correlation statistics, but for the distributions themselves to be visually dissimilar. Thus, we conclude it is important to be cautious about conclusions drawn from a high correlation between various memorisation scores and their proxies, as they may in reality be capturing different properties of data.

## 4 Distillation lowers memorisation

Despite the impressive performance of large neural models on a range of challenging tasks in vision and NLP, practical deployment of such models is often infeasible due to their high inference cost. Recently, *knowledge*

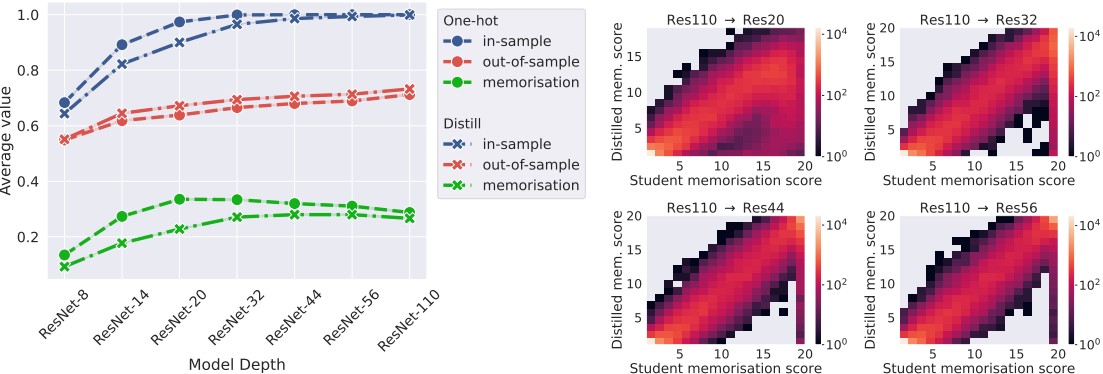

Figure 5: Left: *Average memorisation score is reduced by distillation* across model depths, which can be attributed to both the reduced in-sample accuracy and increased out-of-sample accuracy. Figure shows average memorisation scores from one-hot and distilled models on CIFAR-100 training samples. Right: *Distillation inhibits memorisation of the student*: each plot shows the joint density of memorisation scores under a standard model, and one distilled from a ResNet-110 teacher. As the gap between teacher and student models becomes wider, samples memorised by the student see a sharp decrease in memorisation score under distillation (see the vertical bar at right end of plot). Notably, distillation does not affect other samples as strongly.

*distillation* (Bucilǎ et al., 2006; Hinton et al., 2015) has emerged as a promising approach to compress these large models into more tractable models. Here, one feeds a large ("teacher") model's predicted *distribution* over labels as the prediction targets for a small ("student") model. Compared to the standard training on raw labels (as considered in earlier sections), distillation can provide significant performance gains; these are informally attributed to distillation performing "knowledge transfer". Knowledge distillation has been successfully applied across many applications, including: computer vision (Beyer et al., 2021), language modeling (Sanh et al., 2019), information retrieval (Lin et al., 2021), machine translation (Zhou et al., 2020), and ads recommendation (Anil et al., 2022; Liu et al., 2022).

Compared to standard training, distillation presents us with an interesting setup where models of different sizes interact during the training procedure. This prompts us to ask: how does coupling between models of different sizes during distillation affect, if at all, the memorisation behavior of the resulting student model? Interestingly, while distillation has been shown to yield significant performance gains on average, *training accuracy* has been shown to be systematically harmed (Cho & Hariharan, 2019). Previous work also showed how distillation can lead to worsened *accuracy* on a subset of hard examples (Lukasik et al., 2021). In a related study, model compression has been shown to harm *accuracy* on tail classes (Hooker et al., 2019). These observations indeed hint at the potential impact of distillation procedure on student's memorisation behavior, which we systematically explore in this section.

Towards this, we consider knowledge distillation as conducted using logit matching, and where the teacher is trained on the same sub-sample as the student for estimating the memorisation scores per Equation 2. We provide the hyperparameter details in Appendix D.

**Distillation inhibits memorisation on average.** We begin by investigating what happens to average memorisation under distillation. As discussed earlier, distillation is known to reduce the train accuracy compared to the one-hot models, while increasing the test accuracy (Cho & Hariharan, 2019); in Table 6 (Appendix), we report the train and test accuracies. From this, one could expect distillation to inhibit memorisation. In Figure 5 (left), we illustrate the difference in distributions of memorisation scores across models trained on the ground truth labels (which we call *one-hot training*) and the models distilled from a ResNet-110 teacher model. As expected, we find that distillation tends to *reduce* the number of memorised samples. From the decomposition of memorisation into in-sample and out-sample accuracies for the one-hot and distilled models, we find that the in-sample accuracy becomes lower and out-of-sample accuracy

| Examples set | constant | increasing | decreasing | cap-shaped | other |
|---|---|---|---|---|---|
| All examples | 24.3% | 17.5% | 14.5% | 31.6% | 12.1% |
| Reduced memorisation | 0.0% | 97.2% | 0.0% | 2.3% | 0.5% |

Table 3: Percentages of examples in the CIFAR-100 data as broken down by the trajectory type. The examples where distillation reduces memorisation (as measured by comparing memorisation between the distilled and one-hot trained ResNet-32 models, while using ResNet-110 as the teacher model) are mostly increasing in their trajectory.

becomes higher under distillation. This parallels the observation that train accuracy lowers, and test accuracy increases under distillation.

**Distillation inhibits memorisation of highly memorised examples.** We next turn to analysing the distribution of per-example change in memorisation under distillation. In Figure 5 (right), we report the joint density of memorisation scores under a standard model, and one distilled from a ResNet-110 teacher. We can see that distillation inhibits memorisation particularly for the examples highly memorised by the one-hot model, and especially when the teacher-student gap is wide. Interestingly, none of the examples with small memorisation score from the one-hot model obtain a significant *increase* in memorisation from distillation.

In the Appendix, we also show on per-example trajectories how memorisation lowers especially for the challenging and ambigous examples, which often get high memorisation score value by either the small or large models (Figure 14 and Figure 15; Appendix). In the Appendix, we report further results showing how memorisation is overall lowered across different student and teacher models (Figure 19; Appendix).

**Why does distillation reduce memorisation but improve generalisation?** Our finding is that distillation reduces memorisation but improves generalisation, which is intriguing since recent works suggest memorisation can be beneficial for generalisation (Feldman, 2019). To address this apparent conflict, below we report an analysis of what examples distillation reduces memorisation on. Our hypothesis is that distillation reduces memorisation mostly on hardest examples, and thus releases the generalisation capacity for the model to perform better on other, easier to learn examples. To verify this hypothesis, in Table 3 we provide the breakdown of the CIFAR-100 train data into different categories' counts over the dataset (as defined in Figure 1). In particular, we fix the teacher-student pair of ResNet-110 and ResNet-32 and consider the set of examples S where the memorisation score reduces compared to the one-hot student (one-hot ResNet-32 in the above example). We find that memorisation is mostly reduced by distillation on the examples with an increasing memorisation trajectory examples, which consist of ambiguous and noisy examples (as we note in Section 3.4). Note that this observation relates to the empirical observations about distillation being beneficial in noisy label scenarios (Lukasik et al., 2020).

## 5 Discussion and practical implication

Our study leads to important practical conclusions and avenues for future works. First, one should be careful with using certain statistics as proxies for memorisation. Previous works suggested that various quantities defined based on model training or model inference can provide efficient proxies for memorisation score. Although these proxies appear to yield high correlation to memorisation, we find that distributionally they are significantly different, and do not capture the key aspects of the memorisation behavior of practical models that we uncover. This points at a future direction of identifying reliable memorisation score proxies which can be efficiently computed.

Previous works have categorized the difficulty of examples (Feldman, 2019; Baldock et al., 2021; Jiang et al., 2021a) upon fixing a specific model size. Our analysis points at the importance of characterising examples by taking into account multiple model sizes. For instance, Feldman (2019) refer to examples that have high

memorisation score for a particular architecture as the long tail examples for a given dataset. As our analysis reveals, what is memorised for one model size, may not be for a different size.

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

## A  Limitations

We identify the following limitations of our work:

- While elegant and established as the memorisation score, the metric for memorisation we use (as defined in Equation 1), even with the approximation we use (as shown in Equation 2) is expensive to compute as it requires training of many models. We consider finding an efficient and faithful proxy to memorisation score to be an important future work direction.

- We conduct experiments in the image classification domain. While we do experiment on multiple datasets (CIFAR-100, CIFAR-10, ImageNet, TinyImagenet) and model architectures (ResNet, MobileNet, LeNet), it would be of interest to analyse whether similar trends will be observed in NLP tasks.

## B  Societal impact

We highlight some points regarding the societal impact of our work below:

1. Understanding memorisation is important as it can manifest in undesired behavior by the underlying model. This becomes more critical given that ML models are increasingly being utilized in various areas that affect human life.

2. Even though we have provided a detailed study of model memorisation on standard benchmarks, a domain specific exploration is needed for individual models before they are deployed in practice.

3. As pointed out by our study, the memorisation behaviour of a model is tied to noise in the dataset. Thus, our work emphasizes the need for ensuring data quality.

4. Finally, our memorisation study requires training multiple models which leads to a negative environmental impact. But we hope that the insights from our findings can in the long term serve as guiding principles for future studies and reduce computation in future.

## C  Broader impact

Our work has multiple implications and future work directions.

First, we would like to emphasise the importance of our findings for the broader goal of developing a better understanding of deep learning. Both the increasing bi-modality of memorisation scores with increasing model capacity, and the existence of a variety of trajectories of memorisation scores with increasing model capacity, are to our knowledge new observations in the community. The question of why distillation improves generalisation is still under study by the community. Our work contributes to this branch of work by studying memorisation under distillation.

Next, we identify that the following practical implications can be drawn from our work:

**Our study leads to an important practical conclusion that one should be careful with using certain statistics as proxies for memorisation.** Previous works suggested that certain statistics based on model training or model inference (which we discuss in Section 5) can provide efficient proxies for memorisation score. Although these proxies appear to yield high correlation to memorisation, we find that distributionally they are significantly different, and do not capture the key aspects of memorisation score that we uncover. This points at a future direction of identifying better memorisation score proxies which could be efficiently computed.

**Our study points to a potentially sound way for identifying noisy examples in the labelled data.** Specifically, we find the existence of examples with increasing memorisation over model capacity even after interpolation. We qualitatively find that they are often ambiguous and mislabeled. For example, one general conclusion from this is that if the average memorisation of a subset of train data grows with model capacity, this can point at a possibility of poor quality of the labels in that set.

**Our study points at a potential application of weighting examples based on memorisation score during distillation.** Improving distillation by reweighting or filtering examples is an active research direction in the community. Given the finding how distillation lowers the memorisation of train examples, and how interestingly this is often achieved by lowering the interpolation (as we show in Figure 5), this suggests that memorisation could be used for improving memorisation by filtering out or downweighting examples which the model ends up memorising.

## D   Hyperparameter settings

Our experiments use standard ResNet-v2 (He et al., 2016) and MobileNet-v3 (Howard et al., 2019a) architectures. Specifically, for CIFAR, we consider the CIFAR ResNet-$\{110, 56, 44, 32, 20, 14, 8\}$ family of architectures; for Tiny-ImageNet, we consider the ResNet-$\{152, 101, 50, 34, 18\}$ and the MobileNet-v3 Large architecture with scale factors $\{0.35, 0.50, 0.75, 1.00, 1.25\}$. For all ResNet models, we employ standard augmentations as per He et al. (2016a).

We train all models to minimise the softmax cross-entropy loss via minibatch SGD, with hyperparameter settings per Table 4.

| Parameter | CIFAR-10* | Tiny-ImageNet |
|---|---|---|
| Weight decay | $10^{-4}$ | $5 \cdot 10^{-4}$ |
| Batch size | 1024 | 256 |
| Epochs | 450 | 90 |
| Peak learning rate | 1.0 | 0.1 |
| Learning rate warmup epochs | 15 | 5 |
| Learning rate decay factor | 0.1 | Cosine schedule |
| Learning rate decay epochs | $200, 300, 400$ | N/A |
| Nesterov momentum | 0.9 | 0.9 |
| Distillation weight | 1.0 | 1.0 |
| Distillation temperature | 3.0 | 1.0 |

Table 4: Summary of training hyperparameter settings.

## E   Summary of train and test performance with model depth

In Table 5 we report train and test accuracies across architectures on CIFAR-100 from the one-hot training, while in Table 6 we report train and test accuracies from the distillation training across teachers of varying depths. We find that increasing depth results in models that *interpolate* the training set, while also generalising better on the test set. We also show how how distillation worsens train accuracy while improving the test accuracy.

| Architecture | Train | Test |
|---|---|---|
| ResNet-8 | 0.66 | 0.57 |
| ResNet-14 | 0.82 | 0.65 |
| ResNet-20 | 0.90 | 0.67 |
| ResNet-32 | 0.98 | 0.69 |
| ResNet-44 | 1.00 | 0.71 |
| ResNet-56 | 1.00 | 0.71 |
| ResNet-110 | 1.00 | 0.73 |

Table 5: Train and test accuracies across architectures on CIFAR-100. Consistent with a growing body of work, on the clean dataset we find that increasing depth results in models that *interpolate* the training set, while also generalising better on the test set.

(a) ResNet-110 teacher.

| Architecture | Train | Test |
|---|---|---|
| ResNet-110→ResNet-8 | 0.65 | 0.59 |
| ResNet-110→ResNet-14 | 0.68 | 0.58 |
| ResNet-110→ResNet-20 | 0.78 | 0.64 |
| ResNet-110→ResNet-32 | 0.95 | 0.73 |
| ResNet-110→ResNet-44 | 0.98 | 0.74 |
| ResNet-110→ResNet-56 | 0.99 | 0.75 |

(b) ResNet-56 teacher.

| Architecture | Train | Test |
|---|---|---|
| ResNet-56→ResNet-8 | 0.67 | 0.60 |
| ResNet-56→ResNet-14 | 0.81 | 0.69 |
| ResNet-56→ResNet-20 | 0.88 | 0.71 |
| ResNet-56→ResNet-32 | 0.94 | 0.74 |
| ResNet-56→ResNet-44 | 0.97 | 0.75 |
| ResNet-56→ResNet-56 | 0.98 | 0.75 |

Table 6: Train and test accuracies across architectures on CIFAR-100 under distillation averaged over 5 runs. We find that, compared to the model of corresponding architecture trained using the one-hot objective, distillation lowers train accuracy and in most cases improves test accuracy.

# F   Additional experiments: memorisation score distributions

## F.1   Memorisation score distributions across datasets

In Figure 6 we show the histogram of memorisation scores as a function of model depth. As in the body, we see that increasing depth has the effect of exaggerating the bi-modality of the scores.

## F.2   Memorisation score distributions for varying ResNet width

Figure 7 shows a similar plot on CIFAR-100, where we vary the *width* of a ResNet-32 model.

## F.3   Memorisation score distribution under distillation

Figure 16 shows how memorisation distribution changes under distillation across architectures and datasets. Consistently with the one-hot training depicted in Figure 6 we see both the high memorisation and low memorisation points to increase in number as architecture becomes deeper.

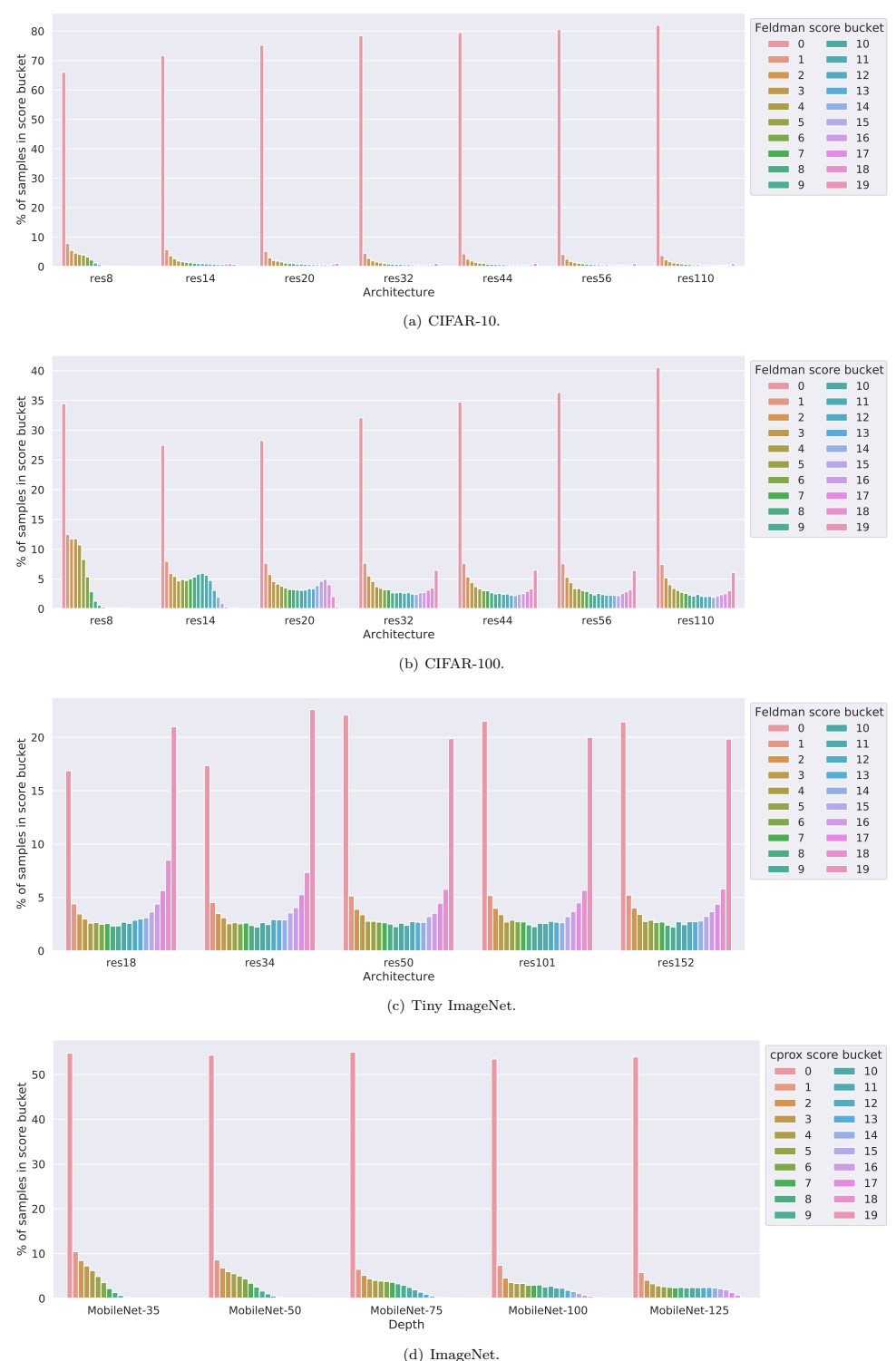

Figure 6: Distributions of memorisation scores across datasets and model sizes. A general trend is that the memorisation scores are *bi-modal* (for sufficiently large models), with most samples' score being close to 0 or 1. Further, this bi-modality is *exaggerated with model depth*: larger models (unsurprisingly) assign low score ("generalise") on relatively more samples, *but* also (more surprisingly) assign high score ("memorise") on relatively more samples as well.

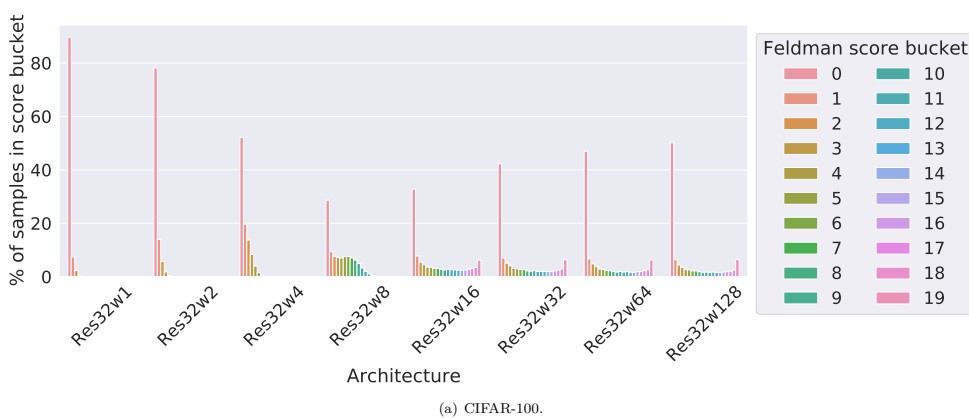

(a) CIFAR-100.

Figure 7: Distributions of memorisation scores across model *widths* of a ResNet-32 on CIFAR-100. A general trend is that the memorisation scores are *bi-modal*, with most samples' score being close to 0 or 1. Further, this bi-modality is *exaggerated with model width*: larger models (unsurprisingly) assign low score ("generalise") on relatively more samples, *but* also (more surprisingly) assign high score ("memorise") on relatively more samples as well.

# G    Additional experiments: larger models memorise some samples *more*

Given the observation of the increasing bi-modality with model depth, we now study how exactly the memorisation score of each example shifts across model depths. In Figure 8, we show how memorisation scores evolve with depth on a *per-example* basis. An unsurprising observation is that memorisation scores in most cases do not significantly change across depths. Beyond this, we can see that as the difference in depths becomes larger, the highly memorised examples according to a small model become *less* memorised by the large model (note the high density in the lower-right part of the plot).

Most interestingly, in all cases, there is a non-trivial fraction of samples whose memorisation score *increases* with model size: note the horizontal bar at the top of multiple subfigures in Figure 8. Increasing memorisation implies a *decreasing* out-of-sample accuracy on such points (as the in-sample accuracy monotonically increases with model depth). This is perhaps surprising, since one expects increasing model depth to improve generalisation (Neyshabur et al., 2019).

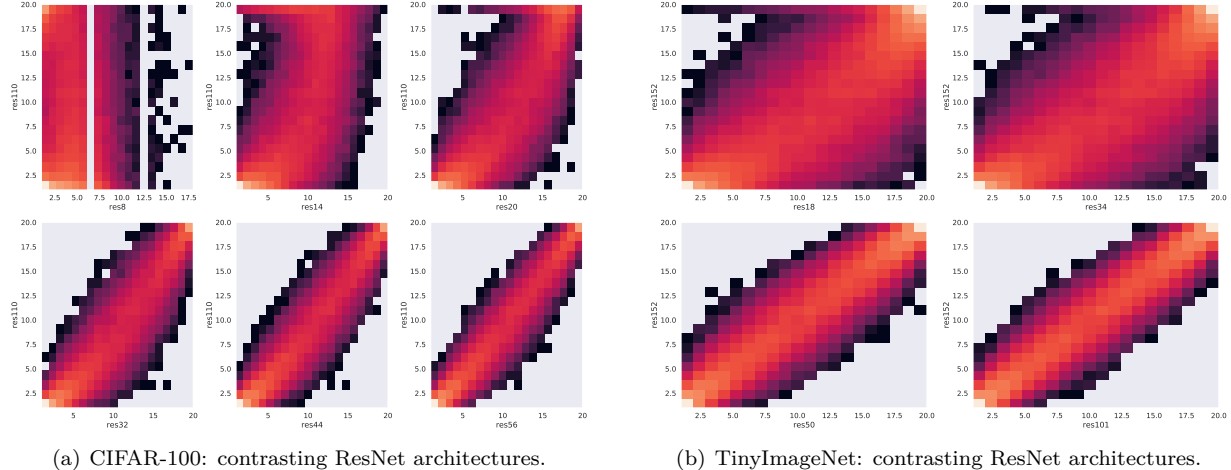

(a) CIFAR-100: contrasting ResNet architectures.          (b) TinyImageNet: contrasting ResNet architectures.

Figure 8: Contrasting per-example memorisation scores across architectures in different setups. As the difference in depths becomes larger, the highly memorised examples according to a small model become *less* memorised by the large model (note the high density in the lower-right part of the plot). On the other hand, smaller models tend to have examples which get assigned a high memorisation score by a large model (note the horizontal bar at the top of a figure).

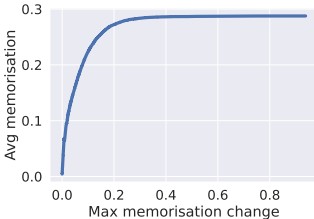

Figure 9: Average memorisation on CIFAR-100 examples where memorisation changes by at most $x$. Least changing examples are less memorised.

## H  Additional experiments and discussion: per-example memorisation trajectories

### H.1  An intuition behind the categorisations.

The categorisations we introduce can be intuitively explained by breaking down the *Feldman memorisation* score into in-sample and out-of-sample accuracy components. For the examples which are easy across model depths, we expect the in-sample and out-of-sample accuracy to be high, and so find a low and unchanging memorisation across depths. Next, for the hard but unambiguous examples, we expect out-of-sample accuracy for that example to be increasing, which corresponds to the gradually increasing generalisation, and in turn, the memorisation score decreasing after reaching the interpolation regime (i.e., where the in-sample accuracy is 100%). This corresponds to the cap-shaped and decreasing trajectories (the interpolation regime is reached even by the smaller models for the latter). Finally, for the mislabeled or ambiguous examples in the data, we find the *correct* label, which disagrees with the *noisy* or ambiguous label in the data, is being recovered better by the deeper models, and so the in-sample accuracy increases or remains high. This corresponds to a high or even increasing memorisation.

Interestingly, Wei et al. (2022) find that noisy examples are eventually memorised by the model (here, memorisation was defined as the confidence in the ground truth label), however the *non-noisy* examples are fitted first. This parallels our observations about memorisation of noisy examples increasing, as the labels get interpolated, but the model does not generalise to the noisy labels.

### H.2  On predicted labels and memorisation trajectories

In Figures 10 and 12 we show additional examples of per example trajectories. We focus on the following three categories: *least changing*, *most increasing* and *most decreasing or U-shaped* in memorisation as the model capacity increases. In Tables 7 and 8 we show predictions from ResNet-20 and ResNet-110 for each example depicted across the Figures. In the discussion below, we refer to examples from Figure 10.

We find that predictions for the *least changing* memorisation examples are saturated in the correct class, demonstrating that these are easy across model architectures. Predictions for the examples with *most decreasing or U-shaped* are assigned low probability from ResNet-20 for the correct class, and high probability from ResNet-110. We can see that these examples are often challenging and mislabeled by reasonable classes early on (e.g., the `bowl` example with a high probability for `clock` and `plate`).

Predictions for the examples with *most increasing* are often assigned high probability from ResNet-20 to categories which are also present in the image (`willow tree` also contains `forest`, and `telephone` also contains `keyboard`). We call these *hard labeled ambigous examples*. The other type we can see are *ambigous examples*, such as `shark` and `pear`, which are not clear and easily confused with other labels as predicted by ResNet-110 (respectively, `dolphin` and `sweet pepper` labels).

We believe the presence of ambiguous examples among both increasing and decreasing memorisation trajectories may be explained by the fact that the multiple labels of ambiguous points may contain labels of various difficulty, with the smaller models assigning high probability to the easy labels, while larger models shift more towards the harder labels. Then, depending on whether the easy or the hard label is present in the training set, different memorisation trajectories are observed. If the observed label is the easier label, as

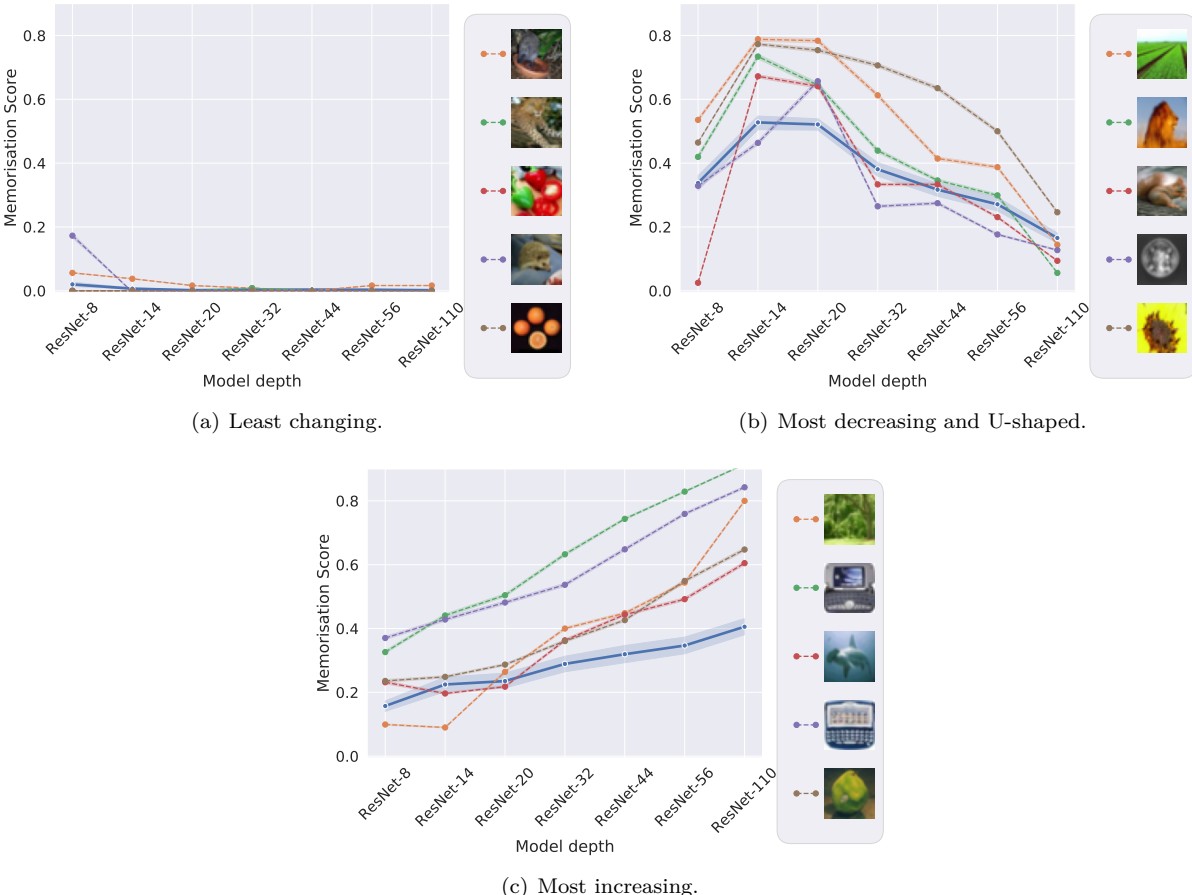

(a) Least changing.

(b) Most decreasing and U-shaped.

(c) Most increasing.

Figure 10: Memorisation of examples across model depths which are interpolated by a ResNet-20 model, and trajectories of memorisation for examples with: the change in memorisation closest to zero, the most decreasing memorisation, and the most increasing memorisation, when comparing ResNet-20 and ResNet-110 architectures. Solid blue line denotes average trajectory according to the top 1% example selected based on the corresponding criterion, and the dashed lines denote top 5 examples according to the corresponding criterion. We find that the fixed memorisation examples are easy and unambigous, as is the case for `porcupine`. The decreasing memorisation example (`plain`) is arguably more complex and gets confused with `caterpillar` and `road` classes by ResNet-20. The increasing memorisation example (`willow tree`) is ambigous (ResNet-110 predicts forest which is arguably also a valid label). In Table 7, we show predictions from ResNet-20 and ResNet-110 for each example depicted across the subfigures.

model size gets larger, more probability gets assigned to the harder labels. Thus, increasing model size may lead to increasing memorisation trajectory. Conversely, if the observed label is the harder label, increasing model size may lead to the decreasing memorisation trajectory.

We note that previous works offered different metrics for identifying difficult examples, with a prominent example of categorisation according to prediction depth (Baldock et al., 2021). We can compare Figure 1 with Figure 29 to see whether the categorisation by memorisation trajectories can be recovered with prediction depth. We see how the most and the least changing examples in terms of their memorisation score are not clearly distinguished when considering prediction depth: most of them get assigned a very high prediction depth scores across architectures.

| Example | Label | ResNet-20 predictions | ResNet-110 predictions |
|---|---|---|---|
|  | porcupine | porcupine: 0.98
crab: 0.01
possum: 0.01 | porcupine: 0.98
shrew: 0.02
girl: 0.00 |
|  | leopard | leopard: 1.00
worm: 0.00
hamster: 0.00 | leopard: 1.00
worm: 0.00
hamster: 0.00 |
|  | sweet pepper | sweet pepper: 1.00
worm: 0.00
girl: 0.00 | sweet pepper: 1.00
worm: 0.00
girl: 0.00 |
|  | porcupine | porcupine: 1.00
worm: 0.00
girl: 0.00 | porcupine: 1.00
worm: 0.00
girl: 0.00 |
|  | orange | orange: 1.00
worm: 0.00
hamster: 0.00 | orange: 1.00
worm: 0.00
hamster: 0.00 |
|  | plain | caterpillar: 0.39
plain: 0.22
worm: 0.20 | plain: 0.86
caterpillar: 0.08
road: 0.02 |
|  | lion | lion: 0.36
camel: 0.16
skyscraper: 0.09 | lion: 0.94
skyscraper: 0.02
hamster: 0.01 |
|  | squirrel | squirrel: 0.36
snail: 0.27
seal: 0.21 | squirrel: 0.91
snail: 0.06
seal: 0.03 |
|  | bowl | bowl: 0.34
clock: 0.29
plate: 0.15 | bowl: 0.87
plate: 0.06
clock: 0.03 |
|  | sunflower | bee: 0.30
sunflower: 0.25
butterfly: 0.20 | sunflower: 0.75
bee: 0.10
butterfly: 0.06 |
|  | willow tree | willow tree: 0.74
forest: 0.21
palm tree: 0.03 | forest: 0.75
willow tree: 0.20
oak tree: 0.02 |
|  | telephone | telephone: 0.50
television: 0.40
keyboard: 0.06 | television: 0.88
telephone: 0.09
keyboard: 0.03 |
|  | shark | shark: 0.78
dolphin: 0.15
whale: 0.05 | shark: 0.40
dolphin: 0.32
whale: 0.21 |
|  | telephone | telephone: 0.52
keyboard: 0.37
clock: 0.08 | keyboard: 0.74
telephone: 0.16
clock: 0.09 |
|  | pear | pear: 0.71
sweet pepper: 0.21
aquarium fish: 0.03 | sweet pepper: 0.45
pear: 0.35
bowl: 0.12 |

Table 7: Predictions from ResNet-20 and ResNet-110 for each example depicted in Figure 10.

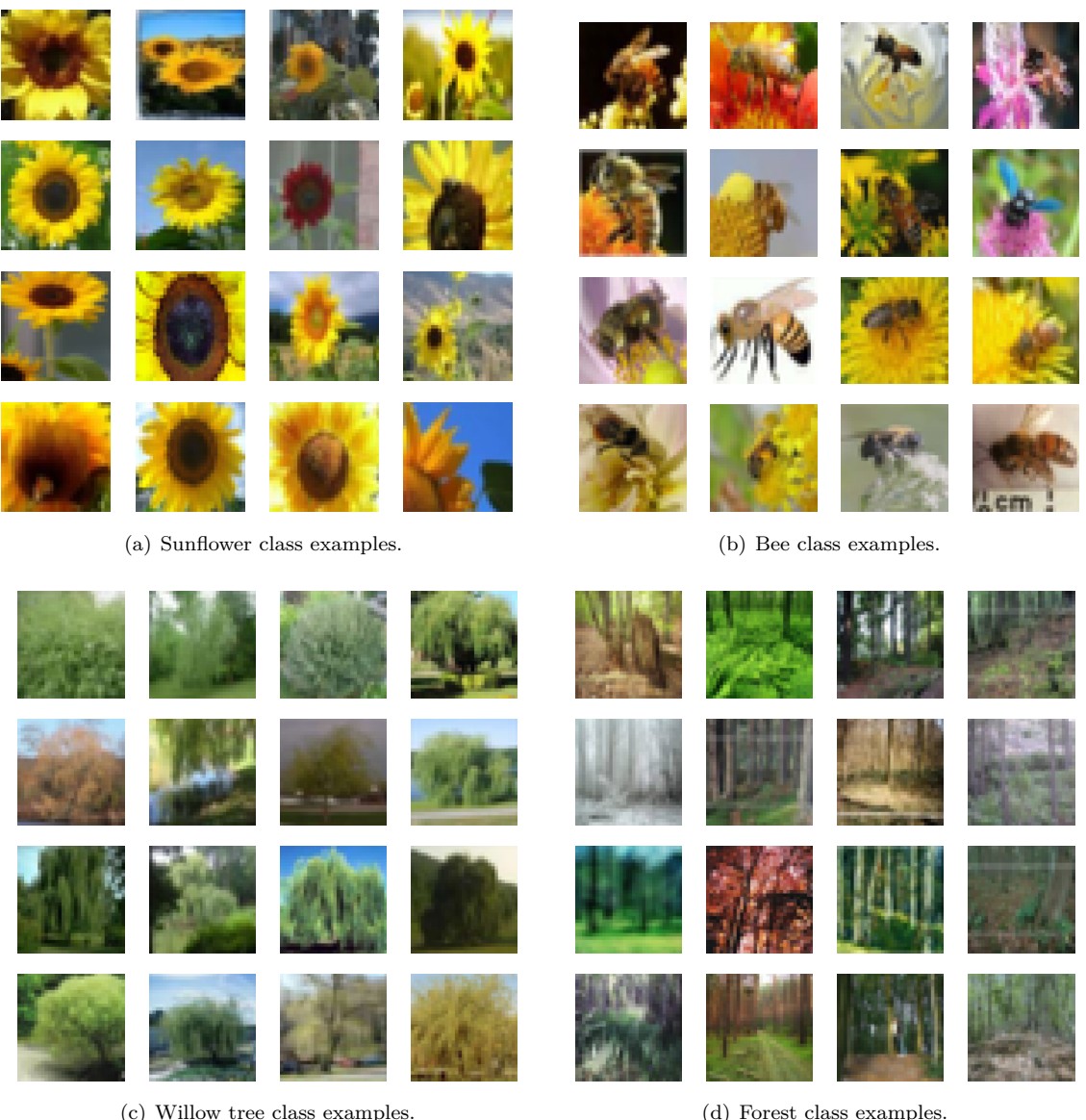

(a) Sunflower class examples.

(b) Bee class examples.

(c) Willow tree class examples.

(d) Forest class examples.

Figure 11: Randomly chosen examples from four classes from CIFAR-100 dataset: sunflower, bee, willow tree and forest.

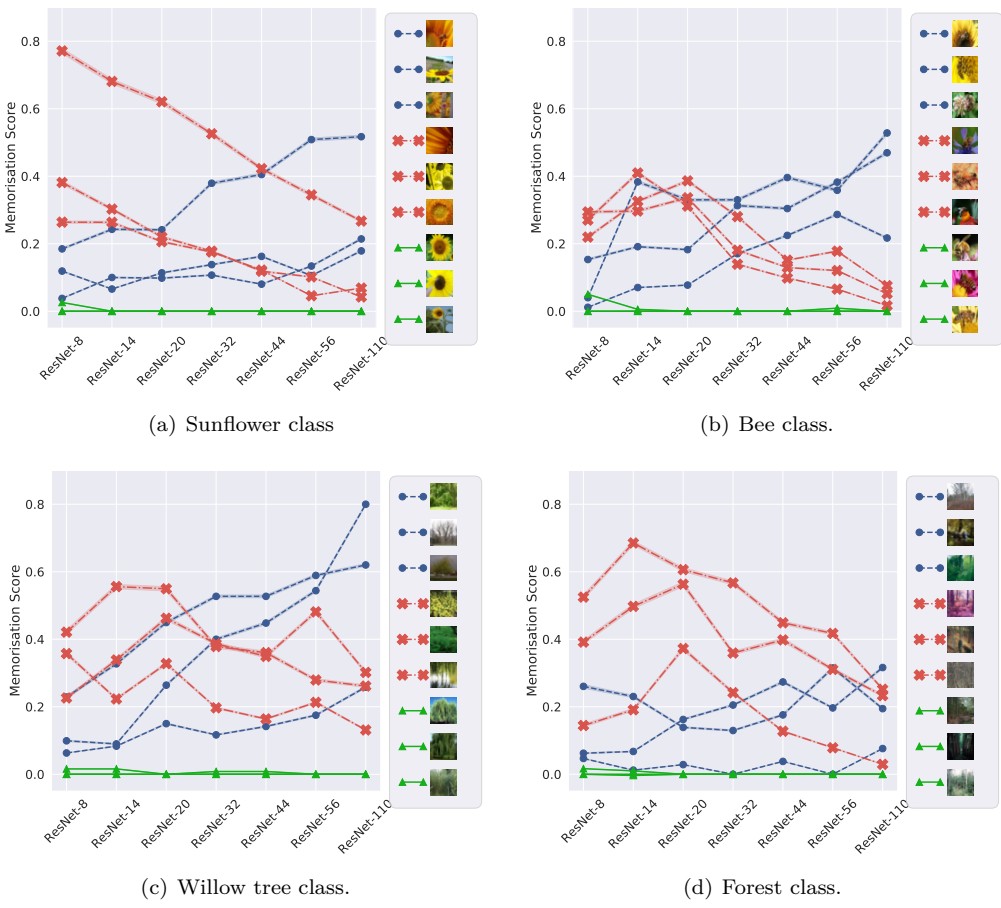

(a) Sunflower class

(b) Bee class.

(c) Willow tree class.

(d) Forest class.

Figure 12: Memorisation of examples from a specific label across model depths which are interpolated by a ResNet-20 model, and trajectories of memorisation for examples with: the change in memorisation closest to zero, the most decreasing memorisation, and the most increasing memorisation, when comparing ResNet-20 and ResNet-110 architectures.

| Example | Label | ResNet-20 predictions | ResNet-110 predictions |
|---|---|---|---|
|  | sunflower | sunflower: 0.76
bee: 0.18
poppy: 0.05 | sunflower: 0.48
bee: 0.45
poppy: 0.06 |
|  | sunflower | television: 0.75
wardrobe: 0.11
can: 0.08 | television: 0.71
wardrobe: 0.12
tiger: 0.10 |
|  | sunflower | willow tree: 0.29
forest: 0.27
maple tree: 0.22 | willow tree: 0.53
forest: 0.41
maple tree: 0.05 |
|  | sunflower | sunflower: 0.38
sweet pepper: 0.20
rose: 0.06 | sunflower: 0.73
sweet pepper: 0.09
pear: 0.03 |
|  | sunflower | sunflower: 0.71
lobster: 0.14
crab: 0.10 | sunflower: 1.00
worm: 0.00
girl: 0.00 |
|  | sunflower | sunflower: 0.70
bicycle: 0.07
spider: 0.05 | sunflower: 0.96
bicycle: 0.02
man: 0.01 |
|  | sunflower | sunflower: 1.00
worm: 0.00
girl: 0.00 | sunflower: 1.00
worm: 0.00
girl: 0.00 |
|  | sunflower | sunflower: 1.00
worm: 0.00
girl: 0.00 | sunflower: 1.00
worm: 0.00
girl: 0.00 |
|  | sunflower | sunflower: 1.00
worm: 0.00
girl: 0.00 | sunflower: 1.00
worm: 0.00
girl: 0.00 |

Table 8: Predictions from ResNet-20 and ResNet-110 for each example from Figure 12.

| Example | Label | ResNet-20 predictions | ResNet-110 predictions |
|---------|-------|----------------------|------------------------|
|  | bee | mouse: 0.31
squirrel: 0.17
possum: 0.14 | mouse: 0.44
squirrel: 0.27
rabbit: 0.08 |
|  | bee | rocket: 0.56
lizard: 0.11
cloud: 0.09 | rocket: 0.44
cloud: 0.21
lizard: 0.09 |
|  | bee | lamp: 0.25
can: 0.23
telephone: 0.19 | can: 0.36
telephone: 0.25
lamp: 0.17 |
|  | bee | bee: 0.70
shrew: 0.15
beetle: 0.05 | bee: 1.00
worm: 0.00
house: 0.00 |
|  | bee | bee: 0.71
wolf: 0.05
tiger: 0.04 | bee: 0.95
wolf: 0.01
shrew: 0.01 |
|  | bee | bee: 0.74
crab: 0.10
beetle: 0.06 | bee: 0.95
cockroach: 0.03
beetle: 0.02 |
|  | bee | bee: 1.00
worm: 0.00
house: 0.00 | bee: 1.00
worm: 0.00
house: 0.00 |
|  | bee | bee: 1.00
worm: 0.00
house: 0.00 | bee: 1.00
worm: 0.00
house: 0.00 |
|  | bee | bee: 0.98
caterpillar: 0.01
squirrel: 0.01 | bee: 0.98
caterpillar: 0.02
worm: 0.00 |

Table 9: Predictions from ResNet-20 and ResNet-110 for each example from Figure 12.

| Example | Label | ResNet-20 predictions | ResNet-110 predictions |
|---|---|---|---|
|  | willow tree | oak tree: 0.65
maple tree: 0.34
pine tree: 0.01 | oak tree: 0.64
maple tree: 0.35
pine tree: 0.01 |
|  | willow tree | oak tree: 0.67
maple tree: 0.32
pine tree: 0.01 | oak tree: 0.87
maple tree: 0.13
hamster: 0.00 |
|  | willow tree | oak tree: 0.37
maple tree: 0.36
pine tree: 0.27 | oak tree: 0.41
maple tree: 0.35
pine tree: 0.24 |
|  | willow tree | willow tree: 0.81
forest: 0.12
pear: 0.02 | willow tree: 0.95
forest: 0.04
bottle: 0.01 |
|  | willow tree | willow tree: 0.46
sunflower: 0.16
forest: 0.16 | willow tree: 0.61
forest: 0.32
caterpillar: 0.03 |
|  | willow tree | willow tree: 0.81
forest: 0.10
pine tree: 0.05 | willow tree: 0.94
forest: 0.05
cloud: 0.01 |
|  | willow tree | willow tree: 0.98
forest: 0.02
worm: 0.00 | willow tree: 0.98
forest: 0.02
worm: 0.00 |
|  | willow tree | willow tree: 1.00
worm: 0.00
hamster: 0.00 | willow tree: 1.00
worm: 0.00
hamster: 0.00 |
|  | willow tree | willow tree: 1.00
worm: 0.00
hamster: 0.00 | willow tree: 1.00
worm: 0.00
hamster: 0.00 |

Table 10: Predictions from ResNet-20 and ResNet-110 for each example from Figure 12.

| Example | Label | ResNet-20 predictions | ResNet-110 predictions |
|---|---|---|---|
|  | forest | road: 1.00
worm: 0.00
girl: 0.00 | road: 1.00
worm: 0.00
girl: 0.00 |
|  | forest | train: 0.76
streetcar: 0.18
pine tree: 0.02 | train: 0.82
streetcar: 0.16
tank: 0.02 |
|  | forest | spider: 0.32
lizard: 0.21
dinosaur: 0.13 | spider: 0.51
lobster: 0.17
table: 0.12 |
|  | forest | forest: 0.80
wardrobe: 0.11
skyscraper: 0.02 | forest: 0.98
wardrobe: 0.02
worm: 0.00 |
|  | forest | forest: 0.86
dinosaur: 0.04
crab: 0.02 | forest: 0.98
bridge: 0.01
tractor: 0.01 |
|  | forest | forest: 0.90
bridge: 0.03
house: 0.03 | forest: 0.99
table: 0.01
worm: 0.00 |
|  | forest | forest: 1.00
worm: 0.00
hamster: 0.00 | forest: 1.00
worm: 0.00
hamster: 0.00 |
|  | forest | forest: 1.00
worm: 0.00
hamster: 0.00 | forest: 1.00
worm: 0.00
hamster: 0.00 |
|  | forest | forest: 1.00
worm: 0.00
hamster: 0.00 | forest: 1.00
worm: 0.00
hamster: 0.00 |

Table 11: Predictions from ResNet-20 and ResNet-110 for each example from Figure 12.

| Example | Label | ResNet-20 predictions | ResNet-110 predictions |
|---|---|---|---|
| | sunflower | sunflower: 1.00
worm: 0.00
girl: 0.00 | sunflower: 1.00
worm: 0.00
girl: 0.00 |
| | sunflower | sunflower: 1.00
worm: 0.00
girl: 0.00 | sunflower: 1.00
worm: 0.00
girl: 0.00 |
| | sunflower | sunflower: 1.00
worm: 0.00
girl: 0.00 | sunflower: 1.00
worm: 0.00
girl: 0.00 |
| | sunflower | bowl: 0.36
sunflower: 0.15
snake: 0.08 | sunflower: 0.61
bowl: 0.11
ray: 0.07 |
| | sunflower | sunflower: 0.84
orchid: 0.08
poppy: 0.02 | sunflower: 0.95
orchid: 0.02
butterfly: 0.01 |
| | sunflower | bee: 0.29
poppy: 0.16
sunflower: 0.15 | sunflower: 0.50
bee: 0.26
poppy: 0.12 |
| | sunflower | sunflower: 0.38
sweet pepper: 0.20
rose: 0.06 | sunflower: 0.73
sweet pepper: 0.09
pear: 0.03 |
| | sunflower | sunflower: 0.71
lobster: 0.14
crab: 0.10 | sunflower: 1.00
worm: 0.00
girl: 0.00 |
| | sunflower | sunflower: 0.78
bowl: 0.09
poppy: 0.07 | sunflower: 0.97
sweet pepper: 0.03
worm: 0.00 |
| | sunflower | sunflower: 0.76
bee: 0.18
poppy: 0.05 | sunflower: 0.48
bee: 0.45
poppy: 0.06 |
| | sunflower | television: 0.75
wardrobe: 0.11
can: 0.08 | television: 0.71
wardrobe: 0.12
tiger: 0.10 |
| | sunflower | willow tree: 0.29
forest: 0.27
maple tree: 0.22 | willow tree: 0.53
forest: 0.41
maple tree: 0.05 |

Table 12: Predictions from ResNet-20 and ResNet-110 for each example from Figure 1.

### H.3 An intuition for why memorisation (mostly) lowers with depth

We have seen that increasing model capacity tends to lower memorisation on average. We have also discussed an intuition behind how this phenomenon is based on the *in-sample accuracy* and the *out-of-sample accuracy*. We next demonstrate a more concrete intuition on a synthetic example for why we expect that should be the case.

We consider a two-dimensional toy binary classification dataset with two classes denoted by black points (class 0) and orange points (class 1), in the form of concentric circles. We construct class 0 in a way that it has a few "outlier" points which are harder to generalize from — here, they are a handful points from an innermost circle in the distribution. In Figure 13, we visualise the decision regions learned by two models of different sizes: a smaller 1 hidden layer model with 10000 dimensions and a larger 4 hidden layers with 500 dimensions. In particular, we visualise the boundaries across multiple random seeds when *excluding* one of the outliers (denoted by the black cross). Concretely, we show the difference between logits returned by the model for the two classes; large positive (negative) values denote areas where the model is confident about the class being positive (negative). The decision boundary (where the logit difference is precisely zero) is denoted in a solid black line.

We observe that the large model learns smooth decision regions, with the inner circle being connected, as opposed to the disconnected areas learnt by the small model. At the same time, when calculating the stability-based memorisation score for the highlighted example, as model size increases, the memorisation score decreases (see Figure 13(c)). This hints that with increasing model capacity, one can expect the model to learn smoother and more contiguous regions; this can lead to better out-of-sample accuracy, and thus lower the memorisation of samples. In I (Appendix) we study a connection of memorisation to robustness using the same illustrative example.

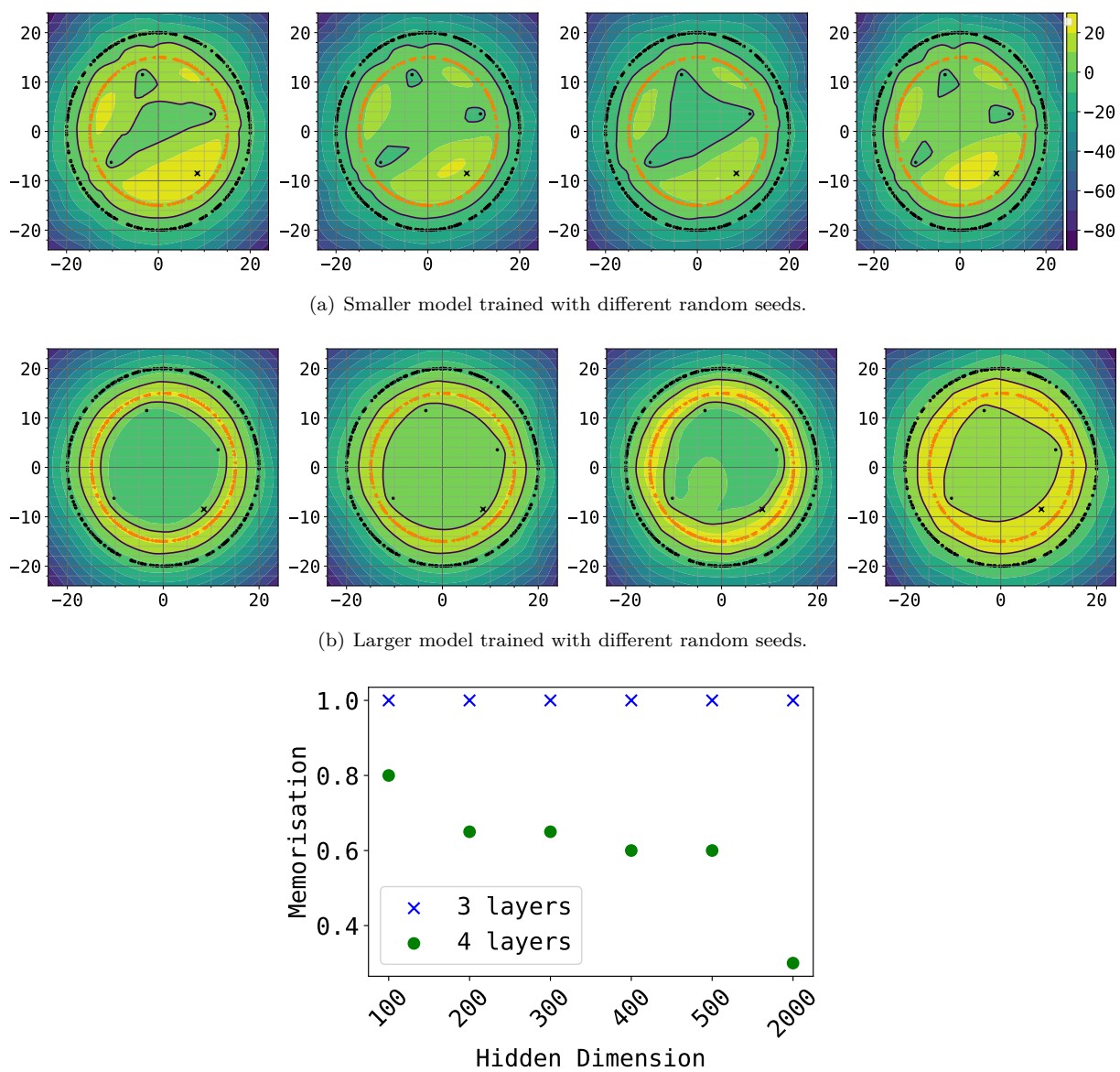

(a) Smaller model trained with different random seeds.

(b) Larger model trained with different random seeds.

(c) Memorisation score trajectory for the outlier example.

Figure 13: *An illustration of an intuition for why larger models on average yield lower memorisation.* We consider a 2D binary classification task where one class is distributed as a single circle (in red-orange), and the other class (in black) is distributed as an exterior circle along with a handful of "outlier" points inside the innermost circle. We visualize the decision boundary (the continuous black lines) when training a smaller model (1 hidden layer model with 10k dimensions; first row) and a larger model (4 hidden layers with 500 dimensions; second row) across multiple random seeds on this task, when *excluding* one of outliers (denoted by the black cross in the bottom-right of the innermost set of points). We observe that the larger model learns decision boundaries that form one contiguous region around the outliers; in contrast, the smaller model learned disconnected decision boundaries around these outliers. This means that for the larger model, out-of-sample accuracy for the excluded outlier is higher; in turn, the memorisation score would be lower. Indeed in the last row, we show the memorisation score for the highlighted example across models with different depths and hidden dimensions. As the model size increases, the memorisation score decreases. This hints that with increasing model capacity, one can expect smoother, contiguous regions learnt by a model and thus lower memorisation of samples.

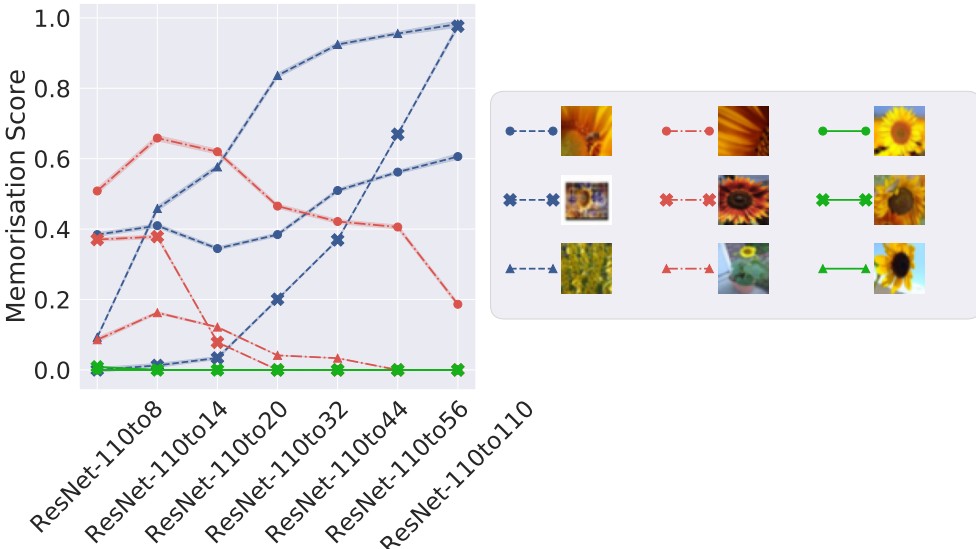

Figure 14: Illustration of how distillation affects how memorisation (in the sense of Equation 1) evolves with ResNet model depth on CIFAR-100 for examples depicted in Figure 1. We find that memorisation is overall lowered for the challenging and ambigous examples.

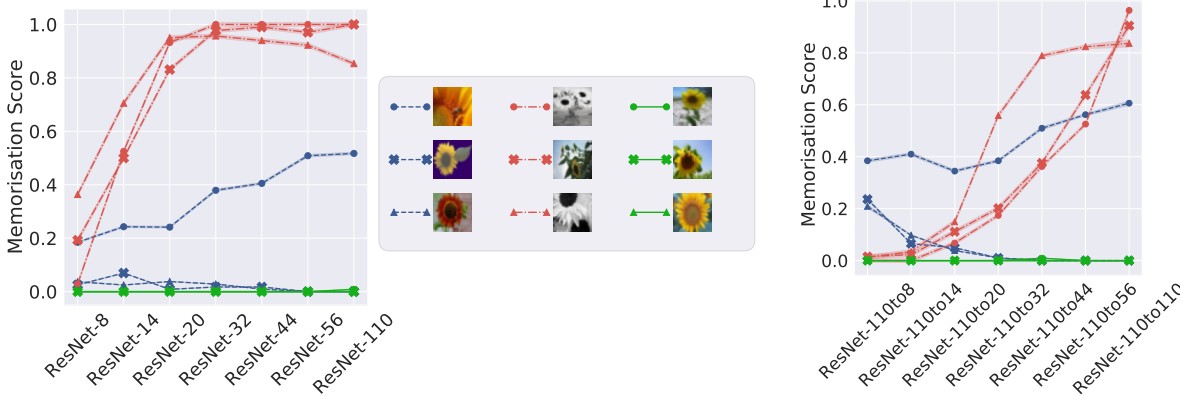

Figure 15: Evolution of memorisation score across one-hot (left) and distilled (right) models for examples across groups: where memorisation is most *increased* by distillation (blue), most *decreased* by distillation (red), and where it *least changes* (green). Distillation reduces memorisation most for hard and ambiguous examples. The remaining groups of examples are easy and unambiguous. A version of this Figure with standard deviations is shown in Figure 20.

## H.4 Memorisation trajectories under distillation

In Figure 14, we plot trajectories of memorisation under distillation for examples depicted in Figure 1. We find that memorisation is overall lowered for the challenging and ambigous examples.

In Figure 16 we report distributions of memorisation under KD. We find that distillation preserves the increasing bimodality of the scores as the student architecture increases.

In Figure 18 we report additional results for how the memorisation scores change under distillation. We confirm the observations from 1(b) that distillation inhibits memorisation, especially for the highly memorised examples.

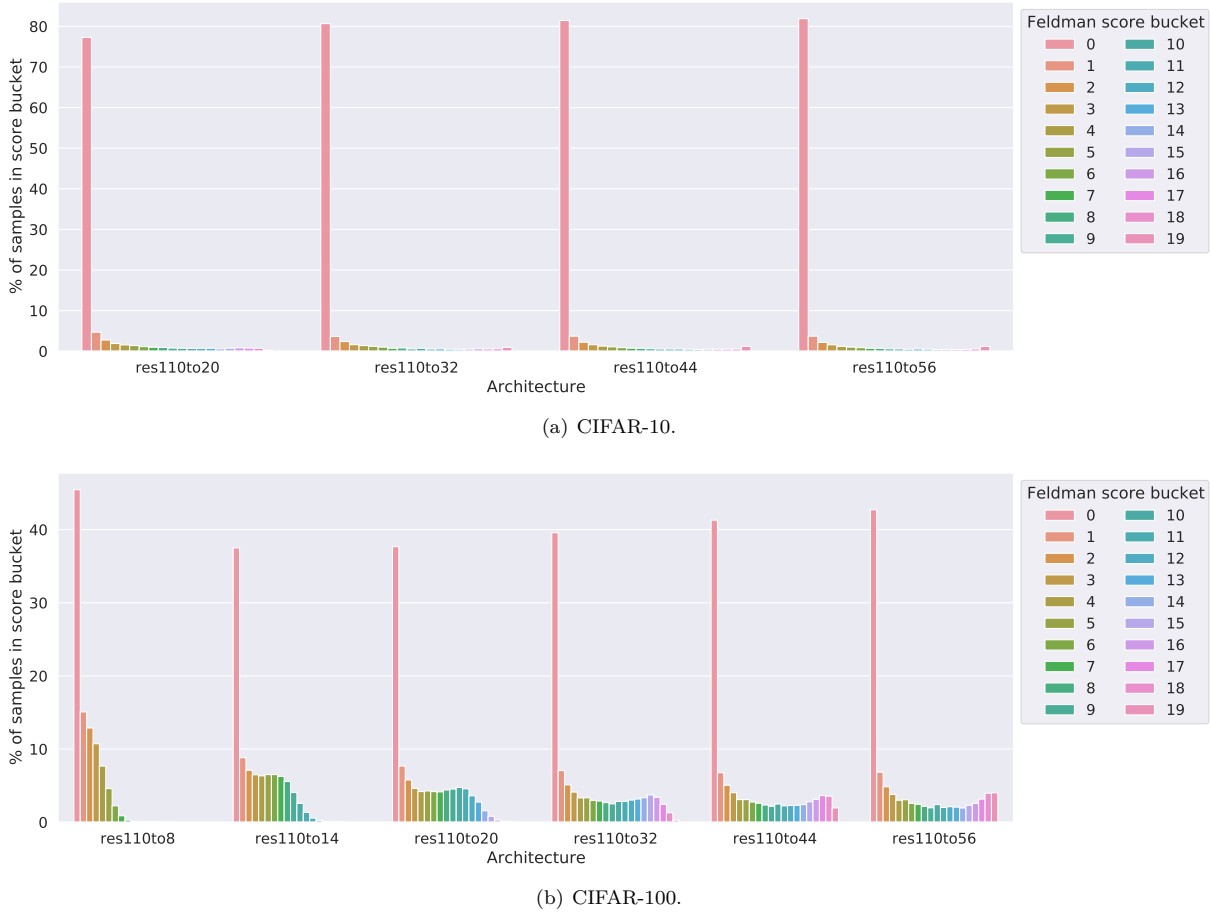

(a) CIFAR-10.

(b) CIFAR-100.

Figure 16: Distributions of memorisation scores under knowledge distillation across datasets (different subfigures) and architectures (different bar colors in each subfigure). Similarly as under one-hot training, memorisation scores are *bi-modal*, and this bi-modality is *exaggerated with model depth*.

Figure 19 studies the effect of varying the teacher model used for distillation. Interestingly, the choice of teacher does not have a strong influence on the results, with even the self-distillation setting for a ResNet-32 resulting in an inhibition of memorisation.

Figure 20 shows the evolution of memorisation scores for examples where memorisation is most increased, most decreased by distillation, and where it least changes. We find how the distillation least affects the memorisation of easy or unambiguous examples.

### H.5 Memorisation trajectories on ImageNet

In Figures 22, 23, 24 we report per example memorisation trajectories for ImageNet with varying MobileNet architecture depths for selected labels: *toaster*, *tent* and *bubble* categories. We make similar observations to those made on CIFAR-100 trajectories. First, the *least changing* trajectories are the easiest examples, with consistently low memorisation across model sizes. Next, both the U-shaped and the decreasing trajectories are composed of harder examples. Finally, the most increasing trajectories are often more ambiguous or with an unclear label (e.g., the first example in the most increasing category for the *tent* label shows a dog), and some of the *bubble* examples don't seem to clearly show bubbles (e.g. the third most increasing example). Thus, we confirm that in the *most increasing* category, the examples can be mislabeled and ambiguous as for their label.

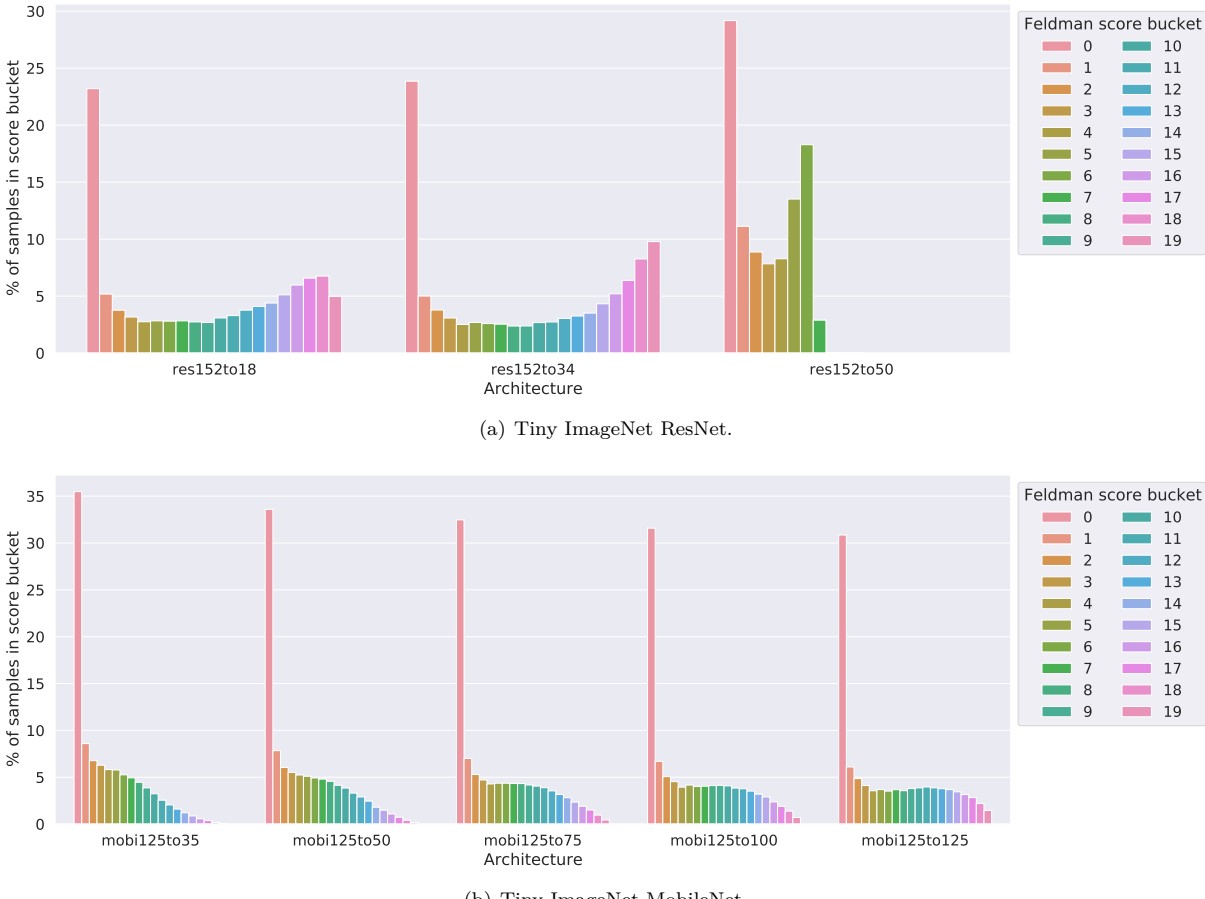

(a) Tiny ImageNet ResNet.

(b) Tiny ImageNet MobileNet.

Figure 17: Distributions of memorisation scores under knowledge distillation across datasets (different subfigures) and architectures (different bar colors in each subfigure). Similarly as under one-hot training, memorisation scores are *bi-modal*, and this bi-modality is *exaggerated with model depth*.

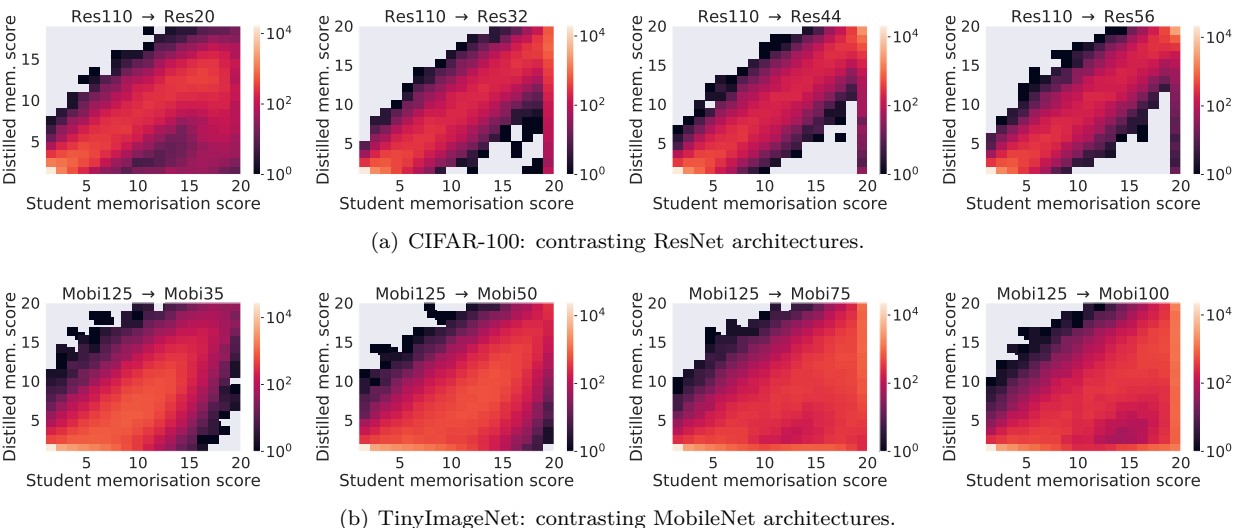

(a) CIFAR-100: contrasting ResNet architectures.

(b) TinyImageNet: contrasting MobileNet architectures.

Figure 18: Contrasting per-example memorisation scores across distilled and one-hot student models in different setups.

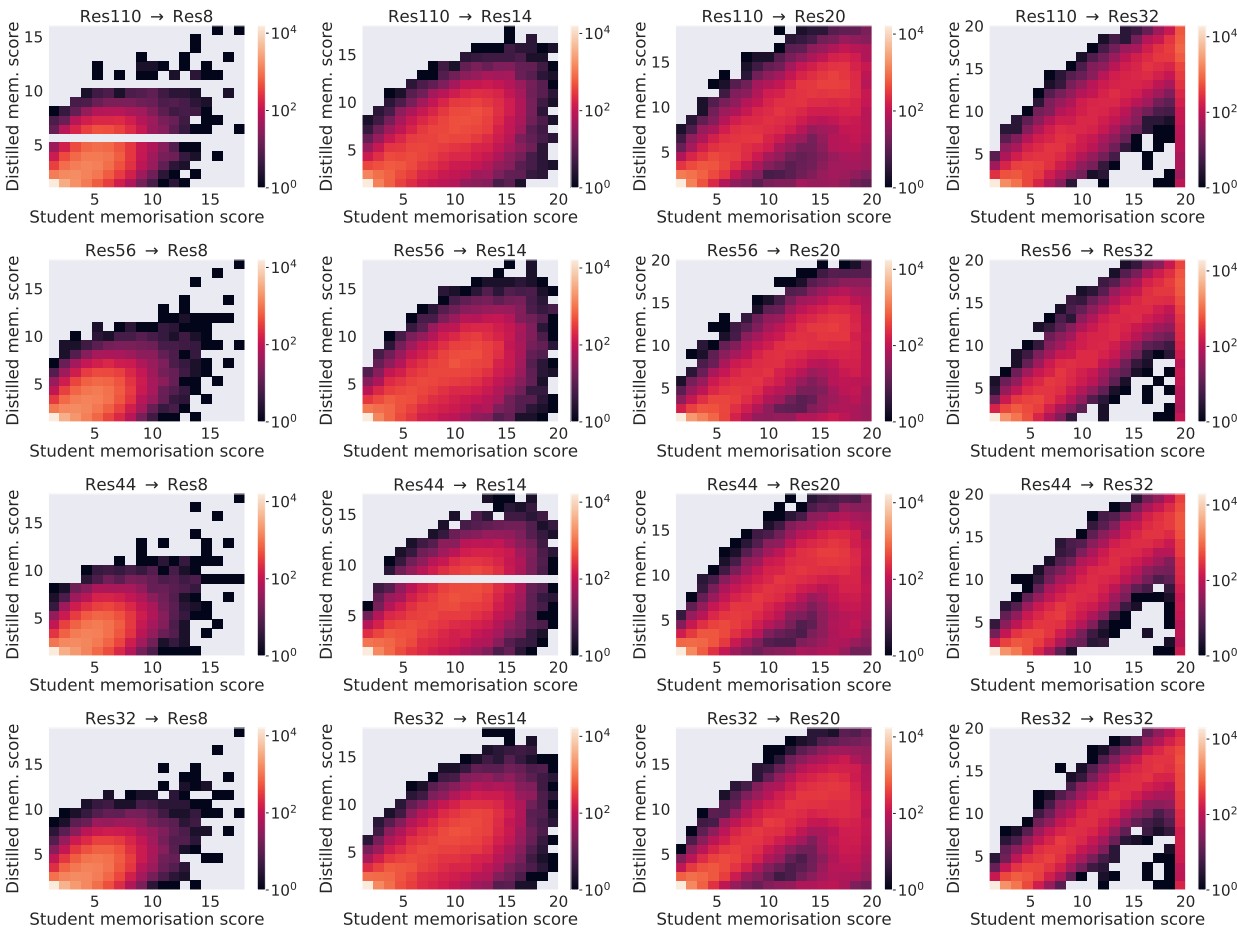

Figure 19: Contrasting per-example memorisation scores across distilled and one-hot student models for various teachers.

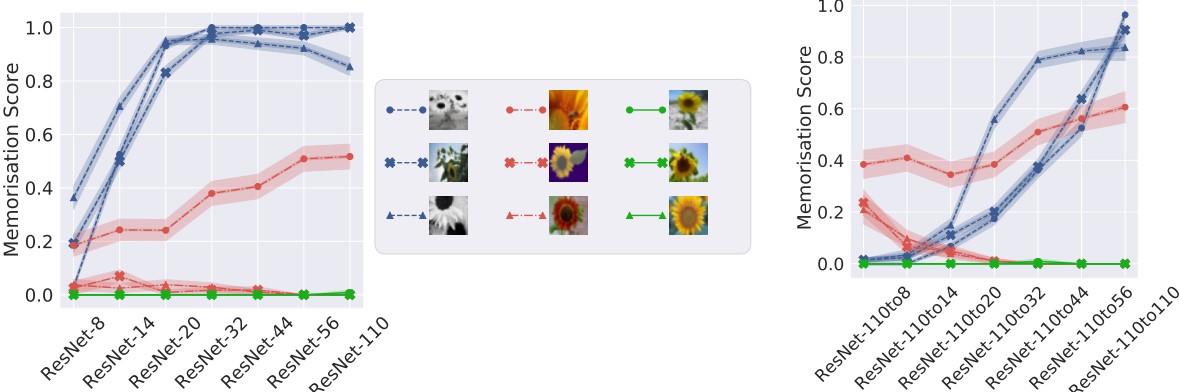

Figure 20: Evolution of memorisation score across one-hot (left) and distilled (right) models for examples across three groups: where memorisation is most *increased* by distillation (blue), most *decreased* by distillation (red), and where it *least changes* (green). We plot standard deviations of the Feldman score estimates around the example trajectories.

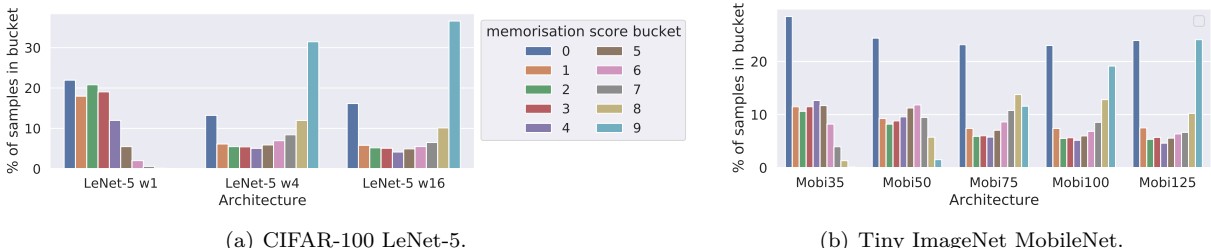

(a) CIFAR-100 LeNet-5.

(b) Tiny ImageNet MobileNet.

Figure 21: Distributions of memorisation scores across datasets and model architectures. We divide the range of memorisation scores into 10 equally-spaced memorisation score buckets. We consider LeNet-5 (LeCun et al., 1998) models on CIFAR-100 training set and MobileNet models on Tiny Imagenet of varying sizes. We achieve multiple architecture versions for LeNet-5 by varying the widths: w$k$ denotes a model where width is scaled $k$ times compared to the original LeNet-5. We find an increasingly *bi-modal* distribution as the width increases.

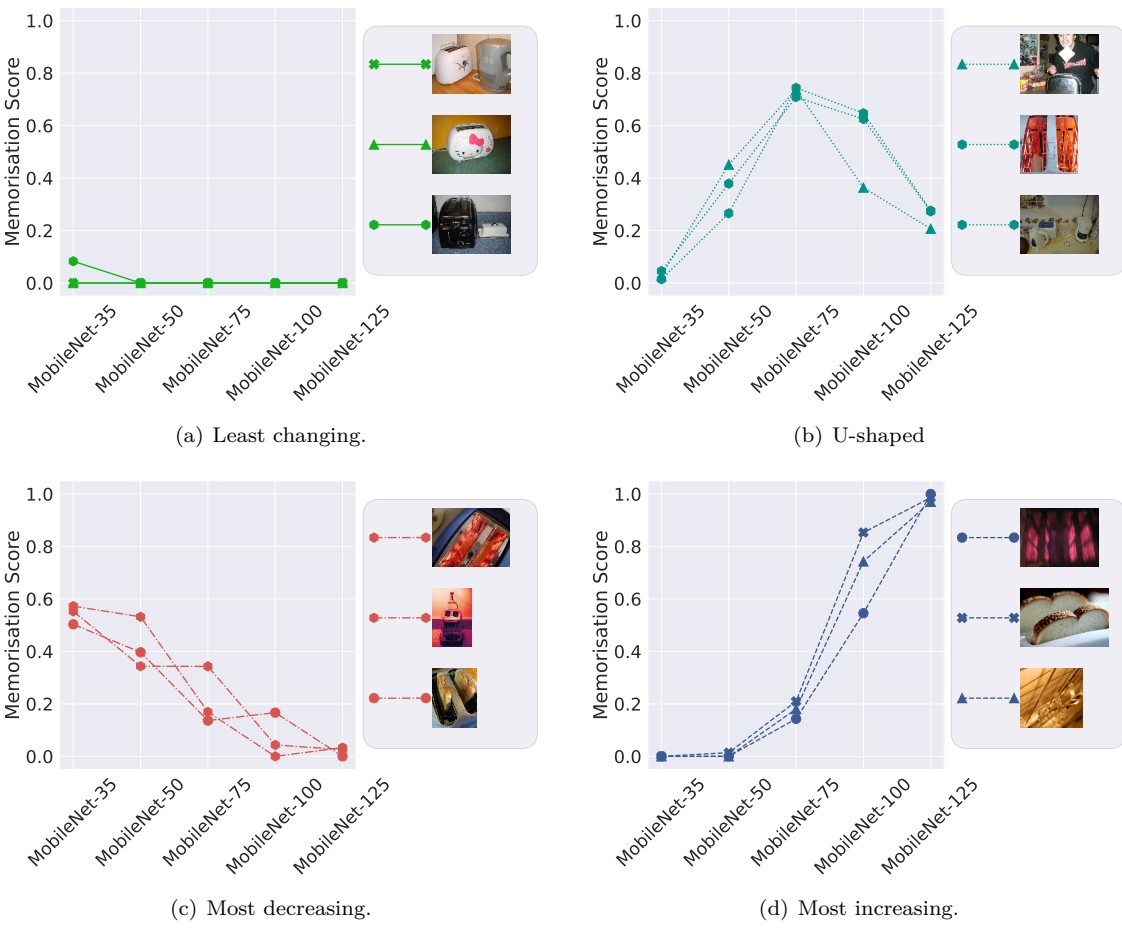

Figure 22: Per-example memorisation trajectories over depth for ImageNet examples from label *toaster*. We show that *training examples exhibit a diverse set of memorisation trajectories* across model depths: fixing attention on training examples belonging to the `sunflower` class, while many examples unsurprisingly have fixed, decreasing or U-shaped memorisation scores (green, red curves), teal curves), there are also examples with increasing memorisation *even after interpolation* (blue curves). Typically, easy and unambiguously labelled examples follow a fixed trend, noisy examples follow an increasing trend, while hard and ambiguously labelled examples follow either an increasing, decreasing or U-shaped trend; in §3.2 we discuss their characteristics in more detail.

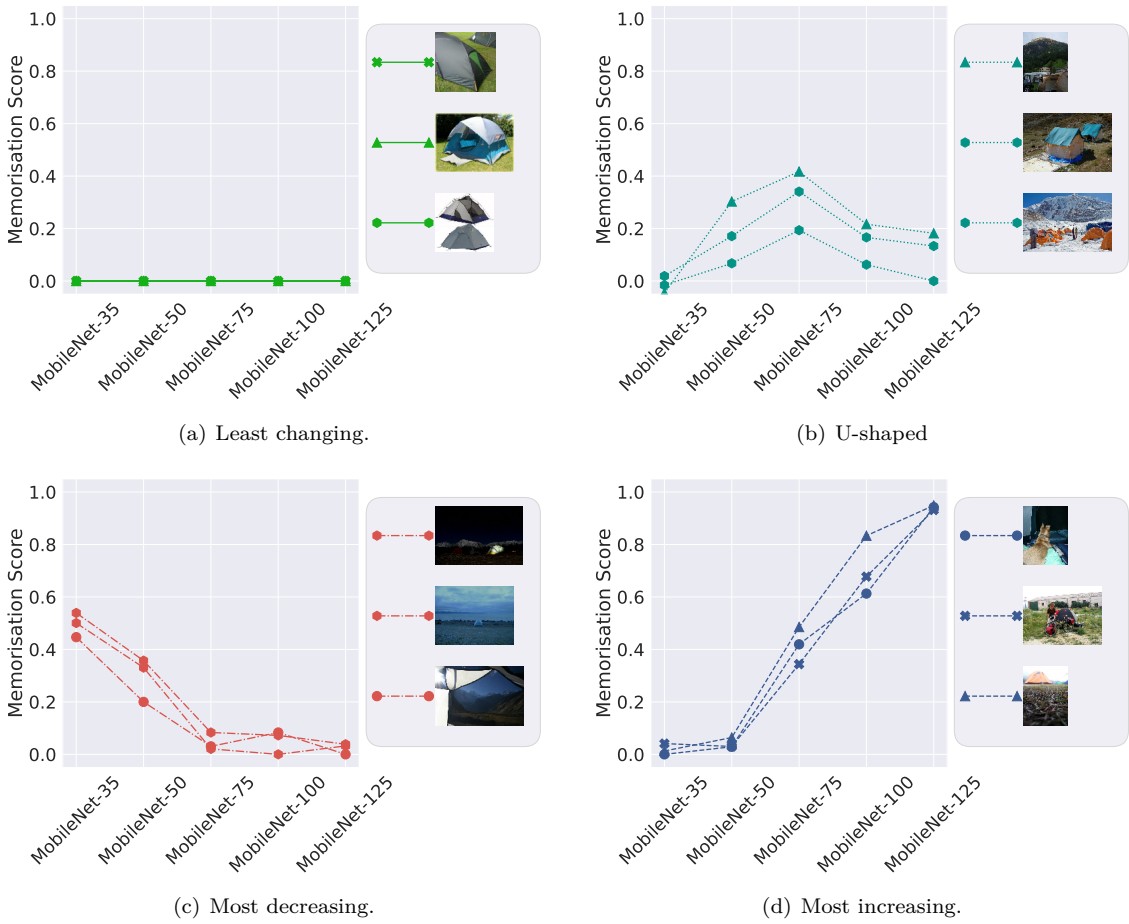

Figure 23: Per-example memorisation trajectories over depth for ImageNet examples from label *tent*. We show that *training examples exhibit a diverse set of memorisation trajectories* across model depths: fixing attention on training examples belonging to the `sunflower` class, while many examples unsurprisingly have fixed, decreasing or U-shaped memorisation scores (green, red curves), teal curves), there are also examples with increasing memorisation *even after interpolation* (blue curves). Typically, easy and unambiguously labelled examples follow a fixed trend, noisy examples follow an increasing trend, while hard and ambiguously labelled examples follow either an increasing, decreasing or U-shaped trend; in §3.2 we discuss their characteristics in more detail.

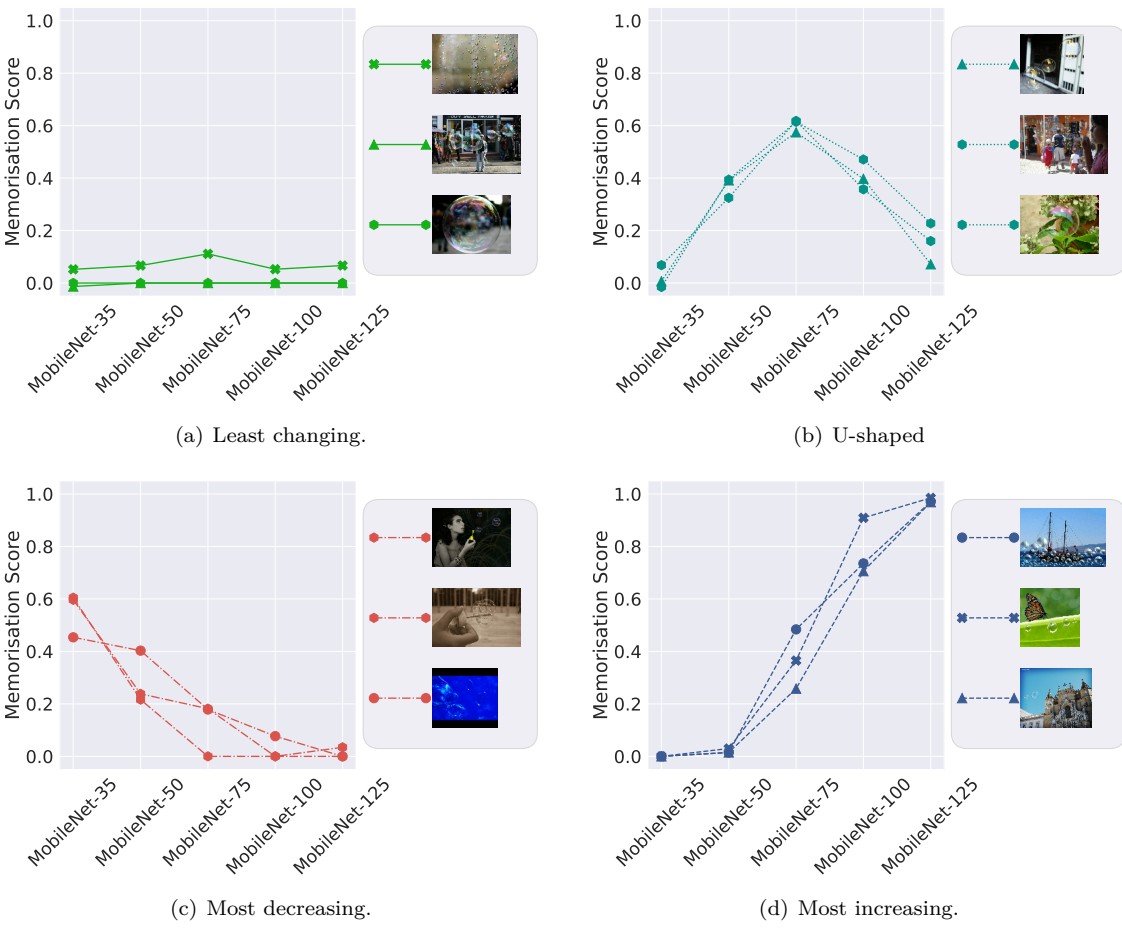

Figure 24: Per-example memorisation trajectories over depth for ImageNet examples from label *bubble*. We show that *training examples exhibit a diverse set of memorisation trajectories* across model depths: fixing attention on training examples belonging to the `sunflower` class, while many examples unsurprisingly have fixed, decreasing or U-shaped memorisation scores (green, red curves), teal curves), there are also examples with increasing memorisation *even after interpolation* (blue curves). Typically, easy and unambiguously labelled examples follow a fixed trend, noisy examples follow an increasing trend, while hard and ambiguously labelled examples follow either an increasing, decreasing or U-shaped trend; in §3.2 we discuss their characteristics in more detail.

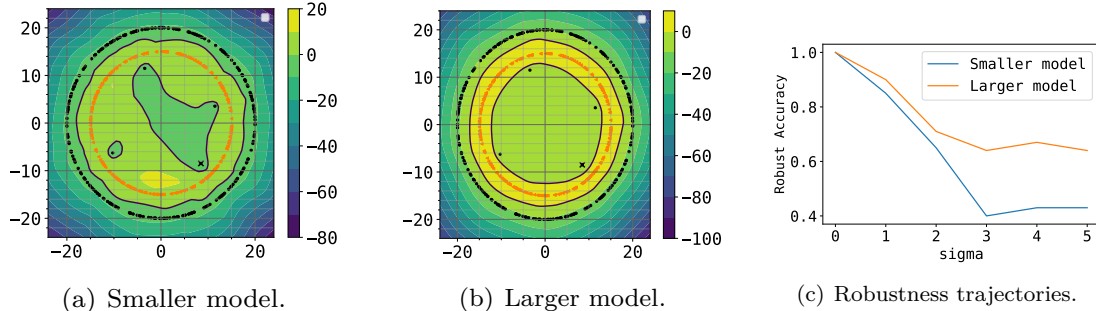

| (a) Smaller model. | (b) Larger model. | (c) Robustness trajectories. |

Figure 25: We show the learnt decision regions (i.e., the difference between the logits from the model assigned to the two classes) when training a smaller (1 hidden layer model with 10k dimensions) and a larger (3 hidden layers with 500 dimensions) models on an illustrative two-dimensional binary classification dataset. Note that unlike in Figure 13, here we *include* the highlighted outlier (denoted by the black cross). We observe how the larger model is more robust than the smaller model on the highlighted outlier, as denoted by the higher accuracy from the larger model on perturbed inputs with random Gaussian corruptions of varying standard deviations (Cohen et al., 2019).

## I   Memorisation and robustness

Intriguingly, a distinct line of work studied the impact of capacity on *adversarial robustness*, e.g. Madry et al. (2018) shows how model robustness increases with capacity. while Papernot et al. (2016) showed how distillation can lead to improved adversarial robustness. It is thus natural to ask how adversarial robustness relates to memorisation. In order to measure the robustness difference between the two models from our example, we revisit the example discussed in Figure 13 and randomly perturb the highlighted outlier with random Gaussian corruptions of varying standard deviations and compare the accuracy on the perturbed example against the original label across the smaller and larger models (Cohen et al., 2019). We plot the decision regions of the compared models and the result of the robustness comparison in Figure 25, and find the larger model to be significantly more robust. That is intuitively expected, as denoted by the learnt decision regions being more smooth, and the inner circle being connected as opposed to the disconnected areas learnt by the small model.

To summarise, on the challenging example, we find the larger model to be more robust, while having lower memorisation.

## J    Additional experiments: prediction depth and cprox

The *prediction depth* has been shown to be closely related to the C-score (Baldock et al., 2021). Prediction depth computes model predictions at intermediate layers, and reports the earliest layer beyond which all predictions are consistent. These predictions were made using a $k$-NN classifier in Baldock et al. (2021). As demonstrated in Baldock et al. (2021), prediction depth forms a lower bound on the stability-based memorisation score.

### J.1    On the choice of probes for prediction depth

Baldock et al. (2021) make a choice between $k$-NN and linear probes. In Appendix E, the authors notice how linear probes lead to all train examples having prediction depth equal to 0, leading to a trivial solution. Therefore, the authors settle on the $k$-NN probes, which don't provide a trivial solution even on the train set. However, the authors consider only three architectures, with the strongest model being ResNet-18, and the largest dataset CIFAR-100.

Our goal is to consider larger scale experiments from Baldock et al. (2021) in terms of both the architectures and datasets, for which $k$-NN probes on large embeddings quickly become computationally prohibitive. In order to allieviate the computational complexity, we consider linear probes on embeddings after average pooling, leading to a non trivial distribution of score values across examples. One supporting argument for this approach is that in ResNet architecture, the embedding after the final layer is subjected to average pooling before passing to the classification layer. Thus, our approach to linear probes is consistent with how ResNet does classification.

In Figure 26, we show the comparison between $k$-NN and linear probes on average pooled embeddings. We see how there is a significant degree of similarity between the two distribution, except the low end of depths where the $k$-NN probes assign more examples. Notice how the distribution of score values for $k$-NN probes (leftmost figure) resembles that from Baldock et al. (2021) (see Figure 1 in Baldock et al. (2021)), despite the latter being computed on non-average pooled embeddings, contrary to the former.

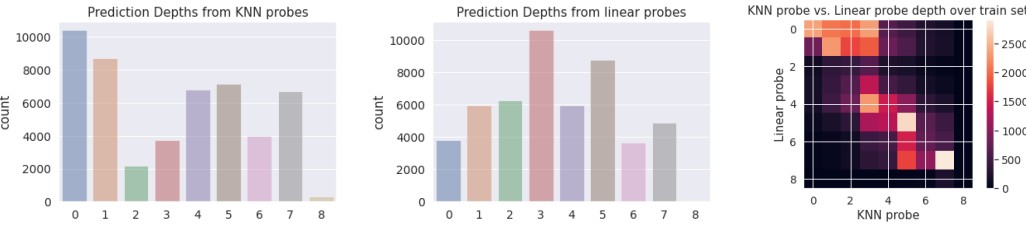

Figure 26: Prediction depth statistics on train examples from CIFAR-10 for ResNet-18 across linear or $k$-NN probes on average pooled embeddings after each layer. Notice how linear probes lead to fewer examples with prediction depth 0 compared to $k$-NN probes (two left-most figures). We notice how there is a high agreement between linear probes and $k$-NN probes (right-most figure), except where $k$-NN probes assign depth 0 or 1, where linear probes increases depth by up to 3, and in the medium range of depths, where linear probes increase the depth by up to 2.

### J.2    Per-sample score trajectories

In Figure 27, we report additional results on how prediction depth and c-score proxy scores change per example across model depths. We confirm that the shift in score values is smaller than that for memorisation score (see Figure 8).

In Figure 28 we provide more details about the prediction depth score distributions. We see how across model architectures, the prediction depth and memorisation scores correlate to a large extent. At the same time, we see that the score marginal distributions are unimodal, contrasting with the bi-modal distributions we found for memorisation score.

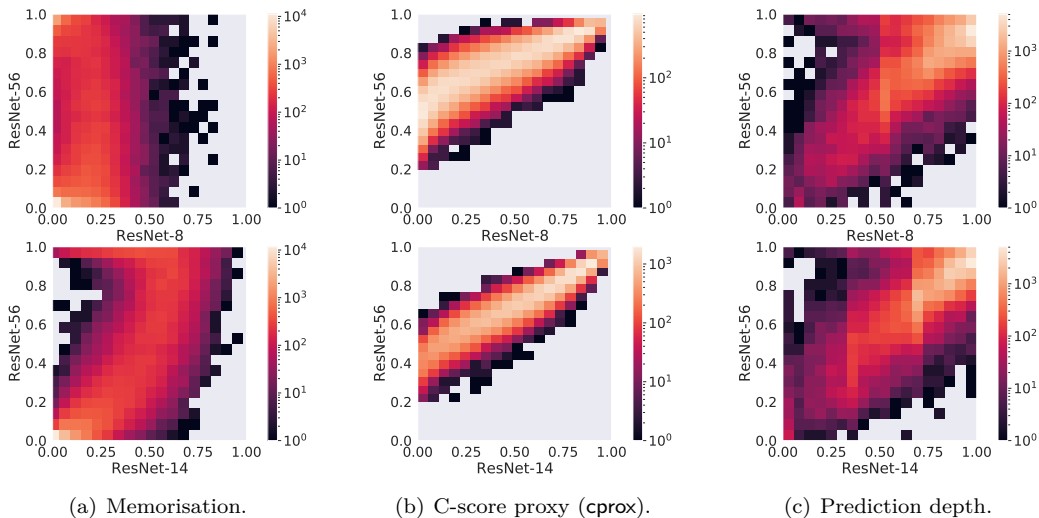

(a) Memorisation.      (b) C-score proxy (cprox).      (c) Prediction depth.

Figure 27: Contrasting per-example stability-based memorisation, prediction depth and cprox scores across architectures in different setups. Notice how cprox plot is concentrated around the diagonal to a greater extent than memorisation score. Prediction depth is also more concentrated around the diagonal, albeit to a lesser extent than cprox (notice there is more mass towards top left part of the heatmap for prediction depth than for cprox, but less than for memorisation score). Overall, the plots suggest that there are relatively few samples whose prediction depth or cprox score changes significantly with increased depth. By contrast, a non-negligible fraction of samples receive a low stability-based memorisation score under a ResNet-8 model, but a much higher score under a ResNet-56 model.

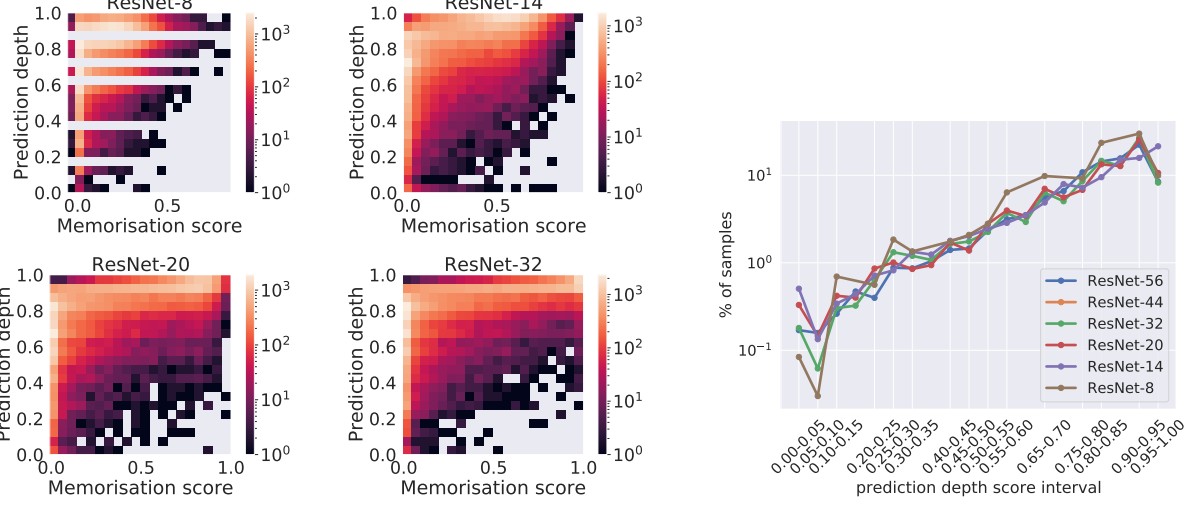

(a) Joint distribution of prediction depth and memorisation.      (b) Marginal distribution of prediction depth.

Figure 28: Relationship between stability-based memorisation (Equation 1) and prediction depth on CIFAR-100. The left plot shows a heatmap of the joint density under the two measures. There is a strong correlation with stability-based memorisation: the correlation to the memorisation score across examples is above 70% for all compared model depths. The right plot shows the marginal distribution of prediction depth over the CIFAR-100 training set across ResNet architectures. The marginals are unimodal, unlike the bi-modal distributions for stability-based memorisation scores.

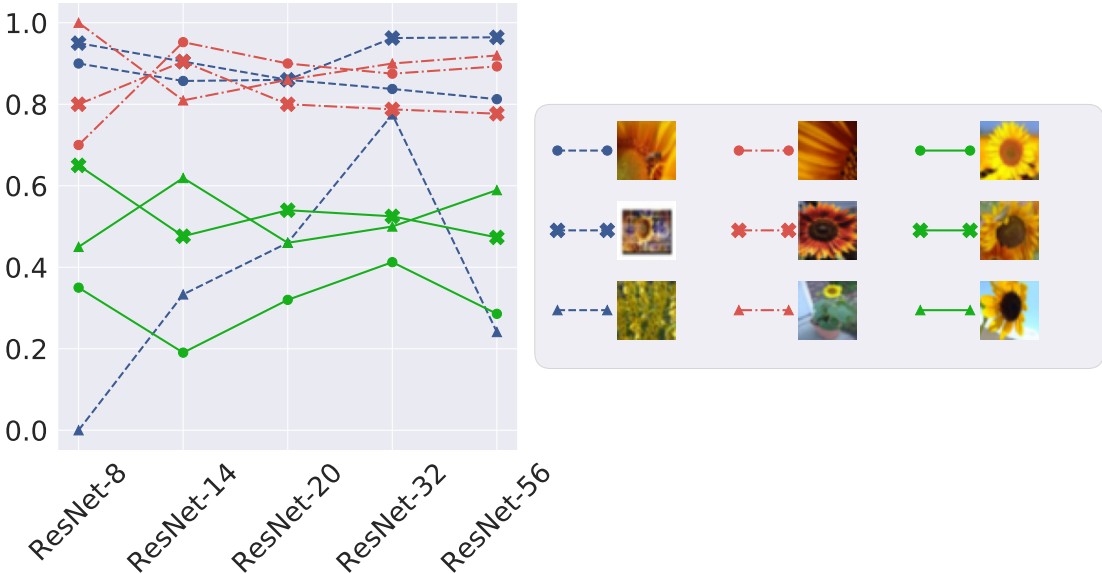

Figure 29: **Example Prediction Depth vs Model Size**. We report the normalized prediction depth score trajectories for the increasing, decreasing and not-changing memorisation examples from Figure 1. We find that examples become less changing in their score when considering prediction depth compared to memorisation score: their trajectory of score over architecture depths is usually roughly constant. This aligns with the heatmap in Figure 27 where see that across various architecture depths the example prediction depth usually doesn't change much.

In Figure 29, we plot trajectories of prediction depth over architecture depths for examples depicted in Figure 1. Recall that in the latter, we found a range of patterns depending on the relative change of memorisation score. For prediction depth, we find that the least changing in memorisation points get the lowest depth across architectures, which may be interpreted as classifying them as the easiest. Interestingly, the most and the least changing examples in terms of their memorisation score are not clearly distinguished between when considering prediction depth: most of them get assigned a very high prediction depth scores across architectures.

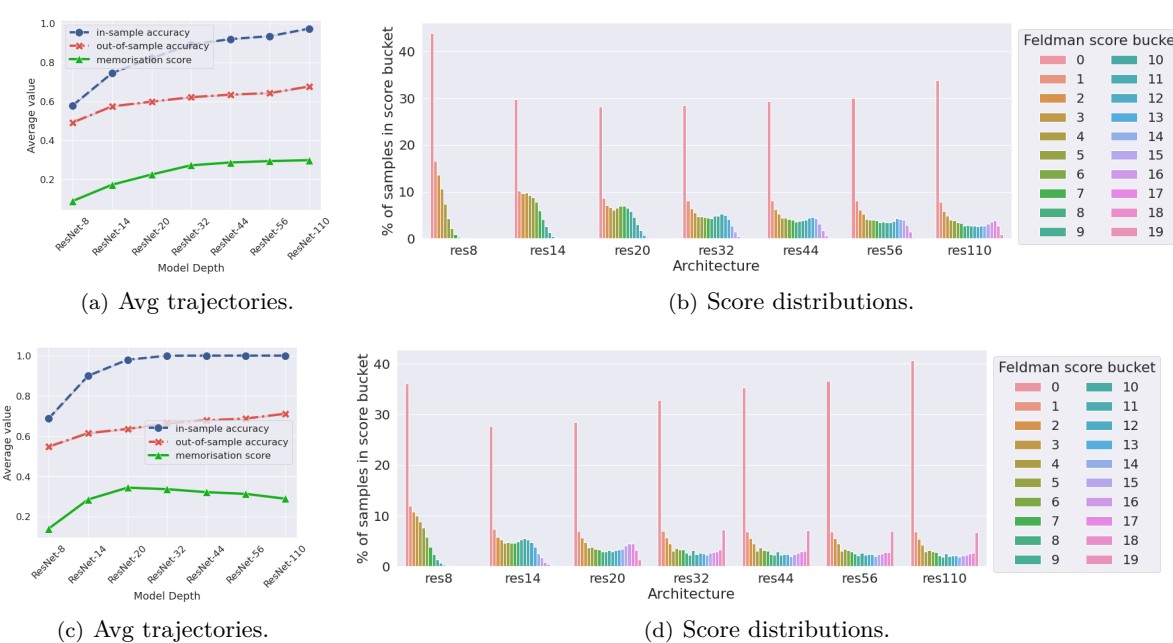

Figure 30: Varying the number of epochs in training ResNet models on CIFAR-100: 300 epochs (top row) and 900 epochs (bottom row). See Figure 2a and Figure 6b in paper for results on 450 epochs. We confirm both the increasing bi-modal score distributions, and the reversed U-shape trajectory of average memorisation. We also notice that as we train for longer, these phenomena become more pronounced. (i.e., there are more clearly bi-modal distributions).

