# OpenReview forum: "What do larger image classifiers memorise?"
_TMLR — Accepted by TMLR_

### Review · Reviewer_Xc8C · 2024-02-06

**Summary Of Contributions:**

This paper makes important contributions to the study of memorization and generalization across image classifier size scales. First, the paper demonstrates that the memorizability of individual points across model scales follows one of a few trajectories: fixed, increasing, decreasing, or cap-shaped. The fixed trajectories correspond to easy points, while other trajectories correspond to harder points. Also, the paper demonstrates that distillation can decrease memorization and improve generalization. The paper also finds that different measures of memorization have different trends.

**Audience:**

Yes

**Broader Impact Concerns:**

None.

**Claims And Evidence:**

Yes

**Requested Changes:**

**Woud strengthen**

Consider Additional Experiments on Settings Outside ResNet on CIFAR-100

Compare the Effects of Increasing Model Scale Along Different Dimensions

Discuss Practical Implications of Distillation-Related Findings

**Strengths And Weaknesses:**

**Strengths**
One of the paper's strengths is its comprehensive empirical analysis. The authors conduct numerous analyses, with many results presented in the appendix. Technically, the results appear novel and significant. As the authors note, the empirical study of how memorization varies with model scale has been understudied. The paper is well-written, and the visuals are effective, especially with the use of clear paragraph headings.

**Weaknesses**
The paper primarily focuses on the ResNet family, and it would be beneficial to include more experiments with other settings, such as the MobileNet architecture on ImageNet. Additionally, the authors only explore one dimension of increasing model scale at a time; it would be interesting to compare the effects of increasing width versus depth, for instance. Finally, the authors make an intriguing finding that distillation can reduce memorization; it would be valuable to discuss in the conclusion how this finding can be practically applied.

---

> ### Author Response · Authors · 2024-03-11
> **Response to Reviewer Xc8C**
>
> We appreciate your thoughtful suggestions. Please see our responses below.
>
> > include more experiments with other settings, such as the MobileNet architecture on ImageNet
>
> Please note **we include MobileNet results on ImageNet in Figure 3**. This corroborates our findings from ResNet and **LeNet** architectures. Overall, our experimental analyses cover multiple datasets (ImageNet, TinyImageNet, CIFAR-100, CIFAR-10) and model architectures (MobileNet, ResNet and LeNet).
>
> > the authors only explore one dimension of increasing model scale at a time; it would be interesting to compare the effects of increasing width versus depth, for instance.
>
> Please note that **we provide the analysis of memorisation across both depth (e.g., Figure 6) and width (e.g., Figure 7) of models**. Notice how our finding of increasing bi-modality of the distribution of memorisation holds for both.
>
> > the authors make an intriguing finding that distillation can reduce memorization; it would be valuable to discuss in the conclusion how this finding can be practically applied.
>
> We agree that exploring new distillation techniques based on our findings would be an exciting direction for future work. One potential direction is to leverage our observation that distillation reduces memorisation mostly on the hardest examples (Table 3). This could motivate an algorithm where distillation helps to identify highly memorised examples, so that such examples can be treated separately from the rest of the training data (e.g. they could be excluded from the next iteration of training or distilling the model).

---

### Review · Reviewer_GZx9 · 2024-02-21

**Summary Of Contributions:**

The paper investigates the impact of model complexity on memorization and generalization in image classifiers, demonstrating that larger models exhibit more bi-modal memorization behaviors, particularly towards ambiguous and mislabeled examples. It highlights how knowledge distillation can mitigate memorization, suggesting its role in enhancing generalization

**Audience:**

Yes

**Broader Impact Concerns:**

none noted.

**Claims And Evidence:**

Yes

**Requested Changes:**

1. please run the models with different initializations and hyperparameters, and see if we can observe similiar behaviors of these initializations.

2. It's unclear why we choose ResNet for cifar100 but MobileNet for ImageNet. It does not seem to me that MobileNet is a standard architecture for ImageNet (or as popular as ResNet). Are there other reasons (other than computing) that lead to this?

3. Overall, this paper is very dense in many topics, it will be better to focus on less topics and discuss more. For example, out-of-sample accuracy does not seem to ever get defined.

**Strengths And Weaknesses:**

Strengths

+ Comprehensive empirical analysis and novel insights into model size versus memorization dynamics.
+ Identifies the nuanced behavior of knowledge distillation in reducing unwanted memorization.

Weakness

- the paper covers a very wide range of topics, but unfortunately didn't cover in-depth of the topics. For example, the discussion can be more interesting if these are presented with wider range of datasets and architectures.
- It's not so clear that how does these affected by initialization and hyperprameters. For example, will Figure 1 be different if we run with different initializations and hyperparameters?

---

> ### Author Response · Authors · 2024-03-11
> **Response to Reviewer GZx9**
>
> We appreciate your thoughtful suggestions. Please see our answers below.
>
> > please run the models with different initializations and hyperparameters, and see if we can observe similiar behaviors of these initializations.
>
> Please note that this is already captured in our experiments.
>
> **Different initializations**: please note that the definition of Feldman memorisation (Equation 1, 2) relies on training many models (in our case, several hundreds of models), each with a different random initialization and over a different training set. In other words, the Feldman score already takes averages over the different initializations.
>
>
> **Different hyperparameters**: for each setup, we rely on standard hyperparameters from previous work (e.g., for our CIFAR ResNet experiments, we follow [He et al., CVPR 2016]).  Thus, our results should capture memorisation as it is likely to be encountered in practical usage of such models. Importantly, since our findings generalize across these **pre-defined** hyperparameter choices, it is reasonable to expect them to be robust to these hyperparameters.
>
> Moreover, we do provide a comprehensive analysis of the role of different choices for the model architecture, including the model widths and depths, across different model architecture families (e.g., ResNet, MobileNet).
>
> Nonetheless, we agree that further ablation of various design choices is of interest. In a **new Figure 30** (Appendix) in the revision, we show results when varying the number of epochs used to train the ResNet models on the CIFAR dataset. Compared to the 450 epochs used for the results in the paper, we run for 300 and 900 epochs and confirm both (a) the increasing bi-modal score distributions, and (b) the reversed U-shape trajectory of **average memorisation**. We also notice that as we train for longer, these phenomena become more pronounced (i.e., there are more clearly bi-modal distributions).
>
> > It's unclear why we choose ResNet for cifar100 but MobileNet for ImageNet. It does not seem to me that MobileNet is a standard architecture for ImageNet (or as popular as ResNet). Are there other reasons (other than computing) that lead to this?
>
>
> Please note that we tried ResNet architecture on TinyImageNet, where we observed similar results as in CIFAR. We also report MobileNet experiments on both TinyImageNet and ImageNet datasets.
>
> Regarding the paper only reporting the MobileNet architecture for ImageNet: our goal was to cover multiple datasets and architectures in the constrained space and under limited computation resources, and so we omitted ResNet on ImageNet for computational reasons (recall that score computation requires hundreds of model runs). See Figures 6, 7, 8, 16, 17, 18, 19 in the Appendix for our additional results across datasets and model architectures.
>
> > out-of-sample accuracy does not seem to ever get defined.
>
>
> Please note we define out-of-sample accuracy in equation (1): see the curly braces under the second component of the right side of the equation.
>
> > Overall, this paper is very dense in many topics, it will be better to focus on less topics and discuss more.
>
>
> We appreciate the reviewer’s comment about the density of our paper. We aimed for a comprehensive study of memorization under different settings. We are open to suggestions about whether anything should be delegated to the appendix.

---

### Review · Reviewer_DrRU · 2024-02-27

**Summary Of Contributions:**

This paper looks at:
* how a memorization metric (memorization score, proposed by Feldman (2019)) changes across model sizes (# of params),
* whether proxy metrics proposed in prior works, which correlate with this quantity, also capture other qualitative attributes (unfortunately not), and
* how knowledge distillation affects memorization (overall less memorization).

It provides empirical evidence using different model architectures on common computer vision classification datasets.

**Audience:**

Yes

**Broader Impact Concerns:**

None - the appendix contains the broader impact statement.

**Claims And Evidence:**

Yes

**Requested Changes:**

1. Potential changes:

Abstract: "While an exciting first glimpse into what real-world models memorise, this leaves open a fundamental question: do larger neural models memorise more?" -> The introduction makes a better point for why we care about this as we do scale up models, so examining this can potentially provide insights for models that are much more expensive to train.

2. Needs clarification:

"Is the improved performance realized by increasing model size merely a result of increased memorisation of training samples by the
larger models, or their improved generalisation?" -> performance is measured on the test set, which is not trained on, so performance should always be independent of memorization. This might require some clarification.

3. Typos:

p.3 "We summarise more notions of memorisation in Table 1 (Appendix)." <- Table 1 is on page 5, not in the appendix.

p. 3 You use an expectation over M to write the consistency score - what is p(M)? This seems wrong or unclear.

p. 5 "A line of work Baldock et al. (2021); Maini et al. (2023); Stephenson et al. (2021)" <- \citep not \citet?

p. 8. Please make Table 2 bigger! Also use percentages (xx.x% instead of 0.xxx).

p. 10: "Why distillation reduces memorisation but improves generalisation?" -> "Why does distillation reduce memorisation but improve generalisation?"

p. 11. Table 3: Also report percentages here please.

**Strengths And Weaknesses:**

The paper is well-written and clear in its exposition. Memorization is of interest. Knowledge distillation is used and researched. It seems it will be of interest to the community by providing a reference for empirical results. The claims seem well evidenced. The paper does not over-promise and is quite cautious with its claims.

---

> ### Author Response · Authors · 2024-03-11
> **Response to Reviewer DrRU**
>
> We appreciate your thoughtful suggestions. We addressed all your suggestions in the revised version of the paper. We think the paper became much clearer now.

---

### Decision · Action_Editor_Tjzu · 2024-03-25

**Recommendation:** Accept as is

**Comment:**

This paper presents an empirical study attempting to answer the question: do larger models memorize more? The authors find that, unexpectedly, increasing model capacity results in a more bimodal distribution on memorization, that this is not always detected by memorization metrics, and that distillation helps avoid memorization. All these findings will be of interest to ML audiences, and since reviewers all agree the authors present enough evidence to support these claims, I recommend acceptance.

**Audience:**

Reviewers unanimously agree that the insights in the paper will be of interest to a subset of TMLR's audience.

**Claims And Evidence:**

Reviewers unanimously agree that the paper presents evidence to support its claims.